# EXPLORE BRIEFLY, THEN DECIDE: MITIGATING LLM OVERTHINKING VIA CUMULATIVE ENTROPY REGULATION

## ABSTRACT

Large Language Models (LLMs) have demonstrated remarkable reasoning abilities on complex problems using long Chain-of-Thought (CoT) reasoning. However, they often suffer from overthinking, meaning generating unnecessarily lengthy reasoning steps for simpler problems. This issue may degrade the efficiency of the models and make them difficult to adapt the reasoning depth to the complexity of problems. To address this, we introduce a novel metric **T**oken **E**ntropy **C**umulative **A**verage (**TECA**), which measures the extent of exploration throughout the reasoning process. We further propose a novel reasoning paradigm—Explore Briefly, Then Decide—with an associated **C**umulative **E**ntropy **R**egulation (**CER**) mechanism. This paradigm leverages TECA to help the model dynamically determine the optimal point to conclude its thought process and provide a final answer, thus achieving efficient reasoning. Experimental results across diverse mathematical benchmarks show that our approach substantially mitigates overthinking without sacrificing problem-solving ability. With our thinking paradigm, the average response length decreases by up to 71% on simpler datasets, demonstrating the effectiveness of our method in creating a more efficient and adaptive reasoning process. Code is available in supplementary material.

## 1 INTRODUCTION

Large Language Models (LLMs) have demonstrated remarkable capabilities in complex problem-solving, particularly when using the Chain-of-Thought (CoT) mechanism. This method breaks down difficult problems into a series of intermediate steps, known as the "thinking" process, enhancing the model's reasoning ability. Further, methods like GSPO, DAPO, and Multi-layer GRPO (Zheng et al., 2025; Yu et al., 2025a; Ding et al., 2025) have been developed specifically to help models tackle increasingly difficult problems with Long CoT, while may raise another significant challenge—overthinking. Overthinking means generate unnecessarily long and redundant reasoning steps, even for simple problems (Cuesta-Ramirez et al., 2025), as an example shown in Figure 1. This not only increases computational costs but can also degrade reasoning accuracy, as models may disregard a correct solution they've already found in favor of further, often incorrect, exploration. This issue makes it difficult for LLMs conduct efficient inference, or adapt the depth of their reasoning to the complexity of the task, therefore has attracted considerable research attention (Sui et al., 2025; Shi et al., 2025; Yue et al., 2025; Xu et al., 2025; Nayab et al., 2024; Huang et al., 2025; Jin et al., 2025; Hong et al., 2025; Yang et al., 2025b).

Reinforcement Learning (RL) is one of the effective approaches to train a reasoning model for better capability(e.g., DeepSeek-R1 (Guo et al., 2025a), DeepSeek-R1-Zero (Guo et al., 2025b), OpenAI o1 (Jaech et al., 2024), QwQ-32B-Preview (Team, 2025; 2024)), while generally the learning focuses on the accuracy reward and format rewards. To tackle the overthinking issue, it is also natural to think about integrating thinking length-related reward into the RL framework (Yue et al., 2025). For example, the thinking length reward assigns higher scores to short, correct solutions while penalizing lengthy or incorrect ones, thereby optimizing the length of the reasoning path. Several existing methods have made various attempts, for example to apply CoT steps, Token numbers or other metrics to build the reward (Sui et al., 2025; Aggarwal & Welleck, 2025; Arora & Zanette, 2025). However, even though simply constraining the CoT length or token count can reduce the output length, they do not necessarily optimize the underlying thinking process. Instead, it may harm essential reasoning steps, forcing the model to prematurely cut short a necessary thought process, which could lead to an incorrect final answer. This creates an unmanaged trade-off between length and accuracy.

To overcome this limitation, we conduct the analysis of the reasoning process, and summarize that the process generally contains two distinct stages: **exploration** and **determination**. The exploration stage involves generating new paths and ideas (Wang et al., 2025b), which is typically marked by high token entropy (representing the model's high uncertainty and the wide range of possible next tokens). The determination stage, on the other hand, is when the

Figure 1: Reasoning process comparison between the original and CER-trained large reasoning models. The original model continues to reflect for four times after the correct answer appears, while the CER-trained model determines the final answer after only one reflection. GREEN: correct answers (first and last shown); RED: reflecting words.

model follows a single, established path (Wang et al., 2025b), with low token entropy. Based on this, we hypothesize that overthinking is caused by unnecessary over-exploration. Building on this insight, we propose an effective metric to indicate the overthinking stage: Token Entropy Cumulative Average (TECA). TECA is a simple cumulative average (Kilic et al., 2023) of all token entropy up untill the current point (Wang et al., 2025b) that represents the model's uncertainty at each inference step. By tracking TECA, we can directly identify the existence of "forking tokens" and, therefore, the exploration stage (Wang et al., 2025b). Our experimental analysis confirm this hypothesis, as shown in Figure. 2, short-thinking models[1] exhibit a TECA that increases briefly at the beginning of a reasoning process and then drops, while long CoT models show a prolonged increase, suggesting excessive and unnecessary exploration[2]. We further propose a novel thinking paradigm: "*Explore Briefly, Then Decide*", which tries to mimic efficient human reasoning, where the goal of problem-solving is to reach a decisive conclusion, not to endlessly explore alternative paths. We apply our TECA into the thinking paradigm by introducing the Cumulative Entropy Regulation (CER) to train an efficient reasoning model within a GRPO reinforcement learning algorithm. CER is able to suppress the model's excessive exploration while preserving its necessary exploration abilities. We further enhance this by using a segmented reward mechanism, applying CER only to correctly predicted samples to improve learning efficiency.

To evaluate our method, we conducted extensive experiments on four math problem benchmarks. The results demonstrate that our approach effectively reduces the response length of models with only a small change in accuracy, as a case shown in Figure. 1. More specifically, our method reduced the response length of Qwen3-4B by 71% on GSM8K and 39.25% on MATH500. For Qwen3-8B, we saw a reduction of 55.21% on GSM8K and 32.76% on MATH500. Our performance also consistently outperforms other existing methods for addressing overthinking. The TECA curves of our trained models visually confirm the desirable "Explore Briefly, Then Decide" thinking pattern, further validating the effectiveness of using TECA as a metric to adaptively adjust the thinking process. By teaching the model to briefly explore and then determine the answer, we enable it to choose the correct extent of exploration, thereby significantly reducing the occurrence of overthinking.

## 2 RELATED WORK

**Overthinking Problem.** DeepSeek R1 demonstrates that long reasoning ability of LLM can be activated by RL(Guo et al., 2025b), however Large Reasoning Models have a severe overthinking problem which manifests as having to think a lot even about simple questions(Wu et al., 2025). For concise reasoning, there are many solutions(Sui et al., 2025). Wang et al. (2025a) found that if the self-reflection tokens such as "Wait" and "Hmm" are suppressed, the response length will decrease up to 27%–51%. Liao et al. (2025)'s work claim that the first reasoning step is crucial. Choosing a correct first reasoning step will reduce the inference cost up to 70%. Moreover, some works start with improving the RL algorithm. Using length-dependent rewards and advantage reweighting can make response length align with the difficulty of the question(Zhang & Zuo, 2025). GFPO(Shrivastava et al., 2025) proposes two metrics: token efficiency and response length for group filtering in GRPO which reduce the response length 46-71% with response length and 71-85% with token efficiency. In addition to modifying the algorithms themselves, there are also works proposing new training paradigms, such as Multi-Stage RL(Tu et al., 2025), D-CoT(Wang, 2025), QFFT(Liu

---

[1] The models don't have the long CoT ability.

[2] Causal diagnostic between TECA trajectories and response length can be found in Appendix A.5

et al., 2025), IBPO(Yu et al., 2025b). Unlike these works, our work uses TECA to analyze the process of reasoning and propose CER which will lead model to learn "Explore Briefly, Then Decide" thinking paradigm.

**Entropy Mechanism.** Information entropy is a concept in information theory which quantifies the average level of uncertainty or information associated with the variable's potential states or possible outcomes(Shannon, 1948). In practical applications of entropy, it is primarily divided into entropy-based optimization methods and entropy-based problem research. Entropy-based optimization add information entropy into the train metrics such as loss, object function. Gao et al. (2025) uses just one single unlabeled carefully selected data to train the model. After only 10 steps optimization, they achieve performance improvements greater than those obtained using thousands of data and carefully designed rewards in rule-based reinforcement learning. Similarly, Agarwal et al. (2025) proposes three kinds of entropy train method: EM-FT, EM-RL and EM-INF. Especially, EM-INF enables Qwen-32B to match or exceed the performance of proprietary models like GPT-4o, Claude 3 Opus, and Gemini 1.5 Pro on the challenging SciCode benchmark which reveals that many pretrained LLMs possess previously underappreciated reasoning capabilities that can be effectively elicited through entropy minimization alone, without any labeled data or even any parameter updates. On the other hand, Wang et al. (2025b) proposed the "forking token" concept which is high-entropy minority token during model answering and such token will lead the model to fork the path and explore more possibility. More important, if gradient updates at the forking tokens instead of all tokens which means stronger desire to explore, the trained model will get higher accuracy on AIME24 and AIME25 contrast to the trained at all tokens. It reveals that the forking tokens make more contribution to answering question correctly. Cui et al. (2025) reveals an inverse relationship between the downstream task performance of large reasoning models trained with reinforcement learning and entropy, and proposes a solution to promote exploration and escape entropy collapse.

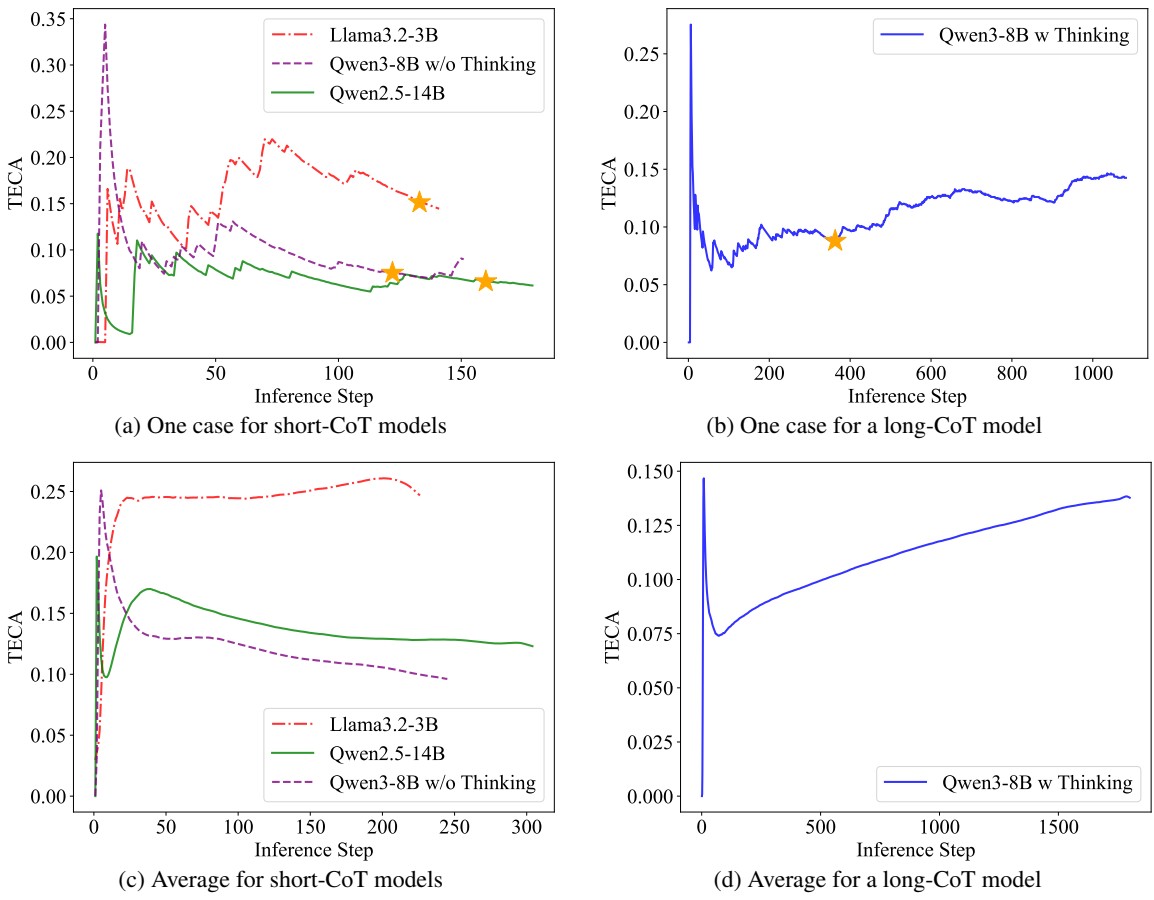

(a) One case for short-CoT models

(b) One case for a long-CoT model

(c) Average for short-CoT models

(d) Average for a long-CoT model

Figure 2: Token Entropy Cumulative Average curves in inference for four compared methods: Llama3.2-3B, Qwen2.5-14B and Qwen3-8B (without and with thinking versions). (a) and (b) correspond to one testing sample and (c) and (d) show the average results of 1000 samples, all from GSM8K dataset. The yellow star marks the step where the correct answer first appears.

## 3 PRELIMINARIES

### 3.1 GROUP RELATIVE POLICY OPTIMIZATION (GRPO)

The GRPO objective extends PPO by optimizing relative advantages within a group of responses and adding a KL penalty to a reference policy, making training more stable and robust to noisy absolute rewards as showed in Equation 1. Specifically, for each question $q$, GRPO uses the old policy $\pi_{\theta_{old}}$ to sample a group of outputs $\{o_1, o_2, , o_G\}$ and then optimizes the policy model $\pi_\theta$ by maximizing the objective.

$$\mathcal{J}_{\text{GRPO}}(\theta) = \mathbb{E}\big[q \sim P(Q), \{o_i\}_{i=1}^{G} \sim \pi_{\theta_{\text{old}}}(O \mid q)\big]$$
$$\frac{1}{G}\sum_{i=1}^{G}\left(\min\left(\frac{\pi_\theta(o_i \mid q)}{\pi_{\theta_{\text{old}}}(o_i \mid q)}A_i,\ \text{clip}\left(\frac{\pi_\theta(o_i \mid q)}{\pi_{\theta_{\text{old}}}(o_i \mid q)}, 1-\epsilon, 1+\epsilon\right)A_i\right) - \beta\,\mathbb{D}_{\text{KL}}(\pi_\theta \,\|\, \pi_{\text{ref}})\right). \tag{1}$$

$$\mathbb{D}_{KL}(\pi_\theta \,\|\, \pi_{\text{ref}}) = \frac{\pi_{\text{ref}}(o_i \mid q)}{\pi_\theta(o_i \mid q)} - \log\frac{\pi_{\text{ref}}(o_i \mid q)}{\pi_\theta(o_i \mid q)} - 1. \tag{2}$$

Here $\epsilon$ and $\beta$ are hyper-parameters, and $A_i$ is the advantage, computed using a group of rewards $\{r_1, r_2, ..., r_G\}$ which are the outputs within each group respectively. The $A_i$ is from the Equation 3.

$$A_i = \frac{r_i - \text{mean}(\{r_1, r_2, \cdots, r_G\})}{\text{std}(\{r_1, r_2, \cdots, r_G\})}. \tag{3}$$

In our work, we use the GRPO algorithm as the base RL train algorithm, and then modify the reward calculation for concise thinking.

### 3.2 TOKEN ENTROPY

The definition of token entropy will follow Wang et al. (2025b). Token entropy is calculated by token generation distribution, independent of specific token. The calculation of token entropy essentially is the information entropy of the event of outputting a token. Here, all possible outcomes of the event correspond to the vocabulary. The formula of token entropy is defined as:

$$H_t := -\sum_{j=1}^{V} p_{t,j} \log p_{t,j}, \quad \text{where } (p_{t,1}, \cdots, p_{t,V}) = \mathbf{p}_t = \pi_\theta(\cdot \mid \mathbf{q}, \mathbf{o}_{<t}) = \text{Softmax}\left(\frac{\mathbf{Z}_t}{T}\right). \tag{4}$$

Here, $\pi_\theta$ represents the LLM parameterized by $\boldsymbol{\theta}$. $\mathbf{Z}_t$, $t$, $V$, $T$ is output logits, inference step(the $t$-th token model generating), vocabulary size, temperature respectively. We choose softmaxed with temperature logits as token probability $p_{t,j}$ which means the probability of model generating the $t$-th inference step, $j$-th token in vocabulary. Then, $H_t$ is sum of all information entropy in vocabulary.

As described in *A Mathematical Theory of Communication* (Shannon, 1948), information entropy can represent the average level of uncertainty. While token entropy formula is based on the information entropy formula, so token entropy represents the uncertainty of next token prediction. **The higher the token entropy, the greater the uncertainty of the model in predicting the next token and vice versa**. This is a significant property about token entropy and LLM generation.

## 4 EXPLORE BRIEFLY, THEN DECIDE: RL WITH CUMULATIVE ENTROPY REGULATION

In this part, we present our Cumulative Entropy Regulation (CER) method for standard GRPO reinforcement learning pipeline. To this end, we first define a new metric Token Entropy Cumulative Average (TECA) and analyze its correlation with overthinking during inference. Based on our hypothesis, we introduce the specific manner (solution) to implement CER with TECA in the RL framework for efficient reasoning.

### 4.1 METRIC: TOKEN ENTROPY CUMULATIVE AVERAGE

According to previous research Wang et al. (2025b) as well as our understanding, in language models, token entropy may describe the uncertainty of next token prediction. In the process of reasoning, it indicates the **exploration extent** at the specific generation step, suggesting the local certainty of the thinking process. However, to optimize the thinking length and meanwhile tackle the overthinking issue, it is required to regulate the whole thinking process instead of single generation step. Therefore, based on token entropy, we propose a new metric—**T**oken **E**ntropy **C**umulative **A**verage (**TECA**)— that may describe exploration extent across the whole thinking process. Specifically, we define the TECA as the average of all of the TE up until the current inference step, specifically as follows:

$$\text{TECA}_t := \frac{\sum_1^t H_t}{t}, \tag{5}$$

where $t$ denotes a specific inference step, also the index of the token generated. $H_t$ denotes the corresponding token entropy referring Eq. 4. The definition explains that at specific inference step $t$, $\text{TECA}_t$ measures the average exploration extent of the thinking process in the past, which indicates the global certainty of the model on the output so far. The higher TECA is, the less certain the model is in one reasoning process.

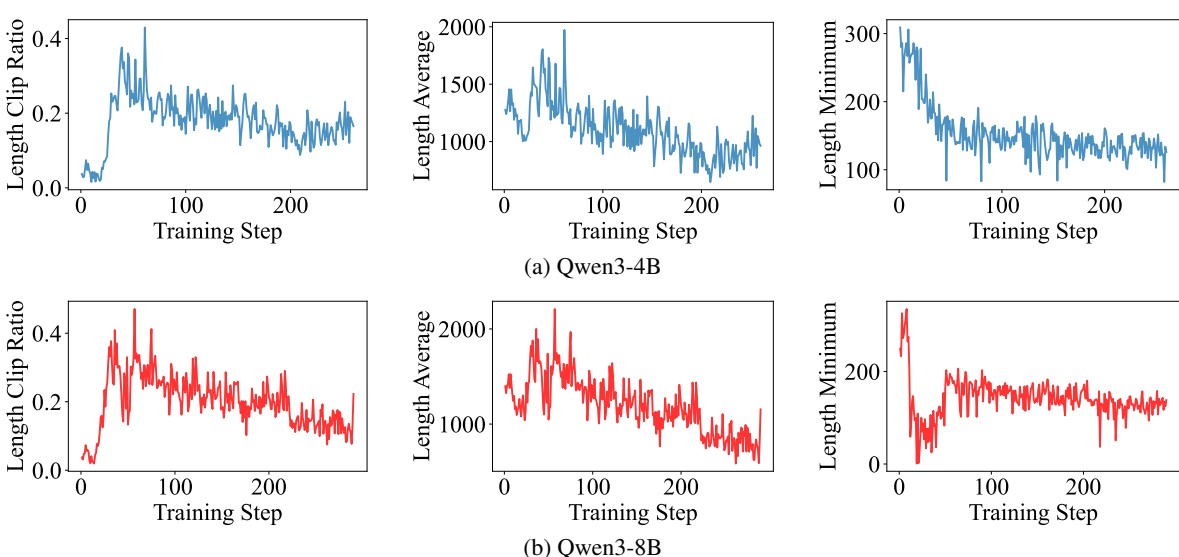

(a) Qwen3-4B

(b) Qwen3-8B

Figure 3: Inference token length curve through CER training on two LLMs. Left: Length Clip Ratio denoting the proportion of responses exceeding the max length of the model; Middle: Average Length of all responses in a group; Right: Minimum Length of all responses in a group.

### 4.2 DISCOVERY: TECA AND OVERTHINKING

To investigate the correlation between TECA and the thinking process, we examined samples from the GSM8K dataset (Cobbe et al., 2021). In Figure. 2, we present the TECA curves for four test settings: one Llama and two Qwen models with thinking enabled or not for Qwen3. We show both a detailed single-sample result and the average results across 1,000 samples for all four test settings, considering only the process with correct answers. For our analysis, we normalized all model responses based on the number of inference steps to align the TECA results across different samples. This was achieved by using first-order linear interpolation to standardize the inference step dimension.

The TECA curves reveal clear patterns during the inference process. At the beginning, all four test settings show a sharp increase or wide fluctuation in TECA, which signals the "**Exploration**" stage. We believe this stage is a necessary part of the model's effort to find possible paths leading to the correct answer. The specific shape of the Exploration stage varies across models, which may be related to their unique training methods. After this initial stage, the TECA for three of the short-thinking settings—*Llama3.2-3B* (Dubey et al., 2024), *Qwen3-8B without Thinking*, and *Qwen2.5-14B* (Yang et al., 2024; Team, 2024)—drops and remains flat (as shown in sub-figures (a) and (c)). This indicates that they have moved into the "**Determination**" stage, as a low TECA suggests the model is no longer generating new thinking paths and is instead focused on confirming the final answer.

However, a different pattern emerges from the *Qwen3-8B* model with thinking mode enabled. Both the single-case and average results show the TECA continuing to rise, suggesting the model keeps exploring without ever entering a determination stage. As illustrated in sub-figure (b), the correct answer often appears very early—at roughly one-third of the way through the inference process—yet the model continues to generate a long, redundant response. This continuous exploration results in an average response length (1897.36 tokens) that is far greater than that of the other models, including *Llama3-8B* (242.113 tokens), *Qwen2.5-14B* (312.153 tokens), and even *Qwen3-8B* without its thinking mode (253.945 tokens). This comparison highlights how an unregulated reasoning process leads to overthinking. Most importantly, our analysis demonstrates a strong correlation between the TECA metric and the model's thinking stages.

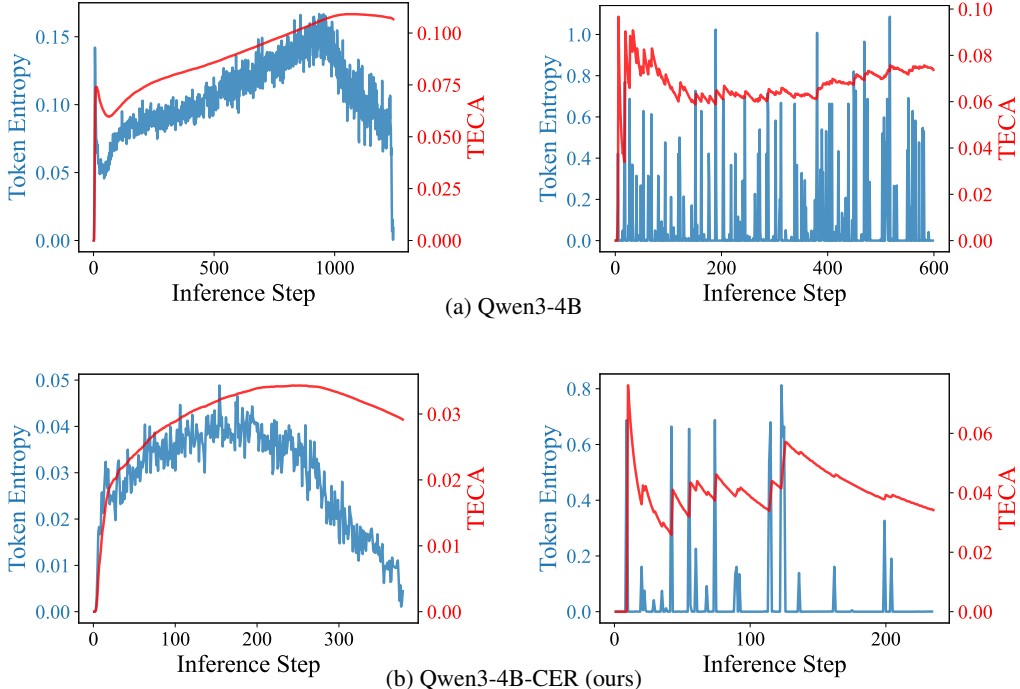

Figure 4: Token Entropy and TECA curves in inference for the reasoning models without and with our CER training. Left: average results for 1000 samples; Right: one case results.

### 4.3 HYPOTHESIS: OVER-EXPLORATION LEADS TO THE OVERTHINKING

As discussed in Section 4.1, higher TECA indicates the uncertainty of the model in the whole reasoning process. Figure 2 illustrates the TECA of *Qwen3-8B* keeping rising during answering, whereas that of *Llama3.2-3B* and *Qwen2.5-14B* going up first, and remaining unchanged or decreasing instead after. Previous study (Wang et al., 2025b) discussed the existence of forking tokens during the reasoning process and pointed out the association between forking tokens with token entropy. Specifically, forking tokens usually correspond to high token entropy. Therefore, we can conclude that our TECA metric may increase when forking tokens appear. On one hand, the presence of forking tokens represents the beginning of model exploration; on the other hand, the continuous increase in TECA indicates the model's uncertainty in its response. Combining these two aspects, we can interpret the model's TECA curve as follows: as reasoning progresses, an increase in TECA indicates that the model is exploring answers, and the response at this stage is filled with uncertainty. A decrease in TECA, however, signifies that the model is becoming more confident in its current response and can output the final, definitive answer.

From the perspective of the model itself, the model attempts to solve this problem by exploration, but the exploration does not lead to certainty about the answer. This causes the model to explore further in an effort to resolve the problem. This process creates a vicious cycle where exploration leads to uncertainty, and uncertainty promotes further exploration, resulting in increasingly lengthy responses, continuous reflection, overthinking, and excessively long answers. Here are the two arguments:

> **Arguments**
>
> - Over-exploration undermines the certainty of a reasoning model on the final answer
> - TECA indicates the overall exploration extent during reasoning, describing the thinking process

Ultimately, abundant exploration originating from the uncertainty leads to lengthy responses, so we propose the hypothesis: **over-exploration leads to overthinking.** To summarize, the higher TECA is, more exploration exist in the reasoning process, the thinking would be more complex. This clearly demonstrates that TECA is effective metric to signal the overthinking issue.

## 4.4 SOLUTION: CUMULATIVE ENTROPY REGULATION ON EXPLORATION

Based on the above arguments and hypothesis, to mitigate the overthinking problem, we propose to regulate the model's exploration during training, but only suppress the model's excessive exploration, not the model's inherent exploratory ability. We expect the model to explore solutions for the problem and then gradually become certain about the final answer after finding the correct path. This process corresponds to the TECA curve first rising and then falling. To achieve this, we designed a TECA reward and a segmented reward mechanism based on the GRPO algorithm. TECA reward is used to suppress the over-exploration and segmented reward mechanism is used to reserve the model's exploratory ability.

### 4.4.1 REWARD FUNCTION

We propose a combined reward function with two components: accuracy and TECA, aiming to suppress the model's excessive exploration and avoid to force the model to output only low-entropy tokens while sacrifice accuracy. We define our TECA reward as follows:

$$r_{\text{te}} = e^{-\text{TECA}_{-1}} + 1, \tag{6}$$

where $\text{TECA}_{-1}$ denotes the TECA value in last inference step during answering. By using $e^{-x}$, the TECA reward is constrained between 0 and 1 aligning with the accuracy reward while maintaining the inverse relationship of TECA with the reasoning efficiency, i.e., thinking length. We apply a trick in the reward function by adding 1 to the TECA reward to encourage the model lean to this part during optimization when multiple reward components are considered. The accuracy reward is defined as:

$$r_{\text{acc}} = \begin{cases} 0 & \text{if } y \neq y_{\text{gt}}, \\ 1 & \text{if } y = y_{\text{gt}}, \end{cases} \tag{7}$$

where $y_{\text{gt}}$ and $y$ denote the ground truth and model prediction respectively.

### 4.4.2 SEGMENTED REWARD MECHANISM

An effective reward should suppress only the model's excessive exploration while ensure its exploratory ability remained. Specifically, the TECA reward is only activated when the model answers correctly, rather than being given regardless of correctness. This ensures that the model receives additional reward only when it successfully answers after sufficient exploration; otherwise, it should consider the accuracy reward only. To this end, we design an effective segmented reward mechanism to delicately combine the two reward components, i.e., accuracy and TECA, as follows:

$$r = \begin{cases} r_{\text{acc}} & \text{if } y \neq y_{\text{gt}} \\ \dfrac{r_{\text{acc}} + r_{\text{te}}}{2} & \text{if } y = y_{\text{gt}}. \end{cases} \tag{8}$$

After the model obtains the correct answer, a lower $\text{TECA}_{-1}$ results in more rewards, thereby encouraging the model to learn to confirm its answer during the "Determination Stage," manifested as a downward trend at the tail of the TECA curve instead of the original trend. We simply average the contribution of the two reward components, i.e., accuracy and TECA, to achieve the combined reward. Note that more sophisticated design could be considered here to integrate multiple components, we leave it for future study. To summarize, with the proposed segmented reward mechanism, reasoning models can learn to autonomously choose the extent of exploration to ensure both answering correctly and obtaining a higher TECA reward.

## 5 EXPERIMENTS

We conduct extensive experiments on two LRMs for the math benchmarks to evaluate the effectiveness of our proposed reward function and learning pipeline. Specifically, we fine-tune Qwen3-4B and Qwen3-8B (Yang et al., 2025a) using

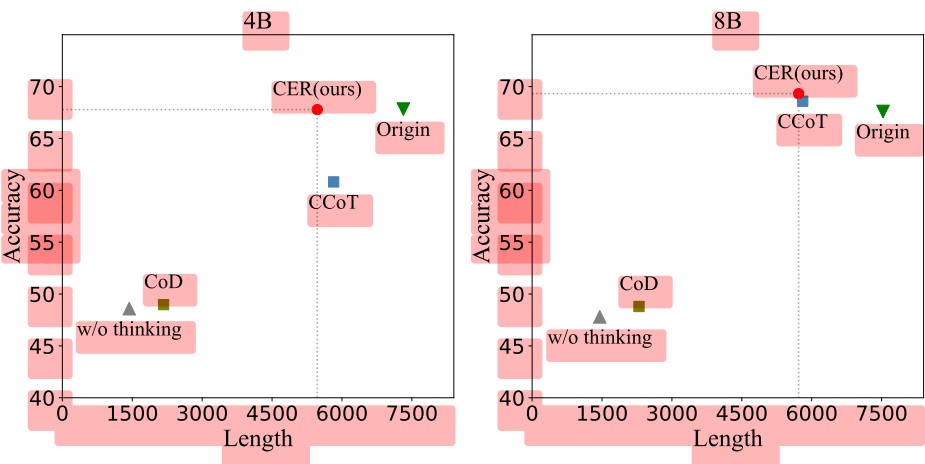

Figure 5: Performance comparison of different thinking methods on the Qwen3-4B and Qwen3-8B backbone models, measured by Accuracy (effectiveness, higher is better) and Token Length (efficiency, lower is better).

LoRA(Hu et al., 2022) by GRPO(Guo et al., 2025b) with the VERL framework(Sheng et al., 2024). Specific hyperparameters can be found in Appendix A.1. We used the GSM8K(Cobbe et al., 2021) training set (7473 samples) as the training data and trained these two models for 5 epochs.

## 5.1 MAIN RESULTS

Figure 3 illustrates the development of the generated token length in inference within the group during the training process. From the figure we can observe continuous dropping patterns for two observed metrics, i.e., Length Clip Ratio and Average Length, showing our learning approach can effectively decrease the token length for average and particularly long thinking process cases. Meanwhile, we can also observe that Minimum Length has a clear lower bound as the curve stays flat along with the training process, suggesting the thinking process will not be unlimited compressed. Such training patterns demonstrate the effectiveness of our CER in terms of suppressing excessive exploration while preserving necessary exploration ability of LLMs in reasoning, thereby tackling the overthinking issue.

The quantitative results on math benchmarks with two models (Qwen3-4B and Qwen3-8B) presented in Table 1 demonstrate the effectiveness and adaptability of our proposed CER method. First, the data shows that CER successfully transforms models prone to overthinking into more efficient reasoning models. Compared to the backbone model, CER significantly reduces the average response length across all four tested datasets, simplifying the thought process without compromising performance. Second, when compared to other existing methods like CoD and CCoT, CER achieves superior accuracy while also reducing response length more effectively. This highlights CER's ability to train models that are not only more efficient but also more accurate, demonstrating its overall effectiveness in a single measure.

Table 1: Performance of different methods on various math benchmarks. Δlen indicates the percentage reduction in the length of the generated answer compared to the origin method (w thinking).

| Method | GSM8K | | | MATH500 | | | AIME24 | | | AIME25 | | | Average | |
|---|---|---|---|---|---|---|---|---|---|---|---|---|---|---|
| | ACC | LEN | ΔLEN | ACC | LEN | ΔLEN | ACC | LEN | ΔLEN | ACC | LEN | ΔLEN | ACC | LEN |
| Qwen3-4B | | | | | | | | | | | | | | |
| w thinking | 92.80 | 1348.59 | - | 65.20 | 4458.60 | - | 64.44 | 11343.57 | - | 48.89 | 12119.62 | - | 67.83 | 7317.59 |
| w/o thinking | 86.50 | 260.96 | 80.65% | 61.20 | 846.35 | 81.02% | 26.67 | 2132.2 | 78.34% | 20.00 | 2503.93 | 82.84% | 48.59 | 1435.86 |
| CoD | 93.30 | 385.50 | 71.41% | 52.60 | 1159.73 | 73.99% | 23.30 | 3607.83 | 68.19% | 26.70 | 3535.23 | 70.83% | 48.98 | 2172.07 |
| CCoT | 82.56 | 616.42 | 54.29% | 64.00 | 2401.94 | 46.13% | 56.67 | 9491.87 | 16.32% | 40.00 | 10775.93 | 11.09% | 60.81 | 5821.54 |
| **CER (ours)** | 94.09 | 391.08 | 71.00% | 64.80 | 2708.65 | 39.25% | 61.11 | 9215.77 | 18.76% | 51.11 | 9565.64 | 21.07% | 67.78 | 5470.29 |
| Qwen3-8B | | | | | | | | | | | | | | |
| w thinking | 94.62 | 1491.38 | - | 65.80 | 4669.74 | - | 63.33 | 11247.68 | - | 46.67 | 12708.16 | - | 67.60 | 7529.24 |
| w/o thinking | 88.86 | 272.02 | 79.83% | 59.00 | 837.16 | 81.22% | 20.00 | 2399.13 | 80.10% | 23.33 | 2300.13 | 80.69% | 47.80 | 1452.11 |
| CoD | 94.40 | 415.80 | 72.12% | 60.80 | 1391.22 | 70.21% | 20.00 | 3657.57 | 67.48% | 20.00 | 3709.20 | 70.81% | 48.80 | 2293.45 |
| CCoT | 92.49 | 739.05 | 50.45% | 65.20 | 2761.19 | 40.87% | 63.33 | 9286.63 | 17.44% | 53.33 | 10438.07 | 17.86% | 68.59 | 5806.23 |
| **CER (ours)** | 92.57 | 668.06 | 55.21% | 65.80 | 3140.04 | 32.76% | 65.56 | 9171.56 | 18.46% | 53.33 | 9894.51 | 22.14% | 69.32 | 5718.54 |

**Case study: The emergence of "reflect only once and then confirm"**

👤 *User:*

**Question:** Janet's ducks lay 16 eggs per day. She eats three for breakfast every morning and bakes muffins for her friends every day with four. She sells the remainder at the farmers' market daily for $2 per fresh duck egg. How much in dollars does she make every day at the farmers' market?

🤖 *Qwen3-8B origin:*

**Response:** <think>Okay, let me try to figure out how much Janet makes at the farmers' market each day. So, first, the problem says that her ducks lay 16 eggs per day.... Hmm, so that's 3 eggs taken away. Then she also bakes muffins for her friends every day using four eggs. Wait, does that mean she uses four eggs for the muffins? So, total eggs used each day would be.... Let me calculate that. 9 times 2 is 18. So, $18 per day? Wait, let me check again. Maybe I missed something.... wait, does bakes muffins with four mean But let me check once more.... Hmm, seems straightforward.... Wait, but.... Yep. I think that's right. So the answer should be $18. </think>To determine how much Janet.... Final Answer $$\boxed{18}$$

**Length:** 969 tokens

🤖 *Qwen3-8B-CER:*

**Response:** <think>Okay, let me try to figure out how much Janet makes everyday at Farmers Market. First, we know that her ducks lay 16 eggs each day. So...Thus, Janet can earn $18 every day at the farmers market. But I should check the steps again to make sure. Confirm: Each day 16 are produced, 7 are consumed (3+4), leaving 9; sold 9×2=$18. Yes, That's correct. </think>Janet's ducks lay 16 eggs per day. She consumes 3 eggs for breakfast and uses 4 eggs for baking muffins, totaling 3+4=7 eggs consumed daily. The remaining eggs available for sale are 16-7=9 eggs. At $2 per egg, her daily income from selling eggs is $9 times 2 = 18$ dollars. $$\boxed{18}$$

**Length:** 385 tokens

Figure 6: A case to compare the responses generated by the original LRM and the CER-trained model. RED: reflecting words; GREEN: correct answer first and last shown.

A particularly important finding is the adaptive nature of CER. As shown in the table, our method achieves a much larger reduction in response length on simpler datasets like GSM8K than on more complex ones, such as the two AIME datasets. This crucial observation confirms that CER does not simply shorten the thought process indiscriminately; instead, it adaptively adjusts the length of the reasoning chain based on the complexity of the problem. This precisely addresses the core issue of "overthinking" by only suppressing unnecessary reasoning steps, validating our central claim that CER promotes efficient, targeted thinking rather than just abbreviated thinking.

## 5.2 THINKING PROCESS ANALYSIS

In Section 4.3, we have explained the correlation between TECA and overthinking, along with the logic for using TECA as an indicator of over-exploration. A primary goal of our CER is to train the model to stabilize its answers after sufficient, but not redundant, exploration. This is the core principle behind our "Explore Briefly, Then Decide" paradigm. To explore whether CER effectively constrains the thinking process and mitigates overthinking, we analyzed token entropy and TECA values across inference steps. The average results from GSM8K test split(1319 samples) and a representative case from Qwen3-4B model are presented in Figure 4. First, by examining the single case results from two models, we can observe that the Qwen3-4B-CER model exhibits fewer high token entropy peaks as the thinking process going. This suggests that the model is generating fewer "forking" tokens and is gradually converging toward a final determination. In contrast, the original Qwen3-4B model shows no clear change in token entropy or TECA during the final stages of the reasoning process, indicating a lack of convergence.

Similar patterns are evident in the average results. The model trained with CER shows a clear increase in TECA at the beginning of the inference, indicating a deliberate and considerable exploration phase, suggesting our CER does not harm the critical exploration ability of the model. More importantly, we can observe over-exploration is effectively suppressed, as evidenced by a clear drop in TECA in the latter part of the inference. This demonstrates that the thinking process is reaching a determination stage and is actively avoiding overthinking. This behavior directly supports our claim that CER specifically targets and suppresses only the unnecessary "overthinking" portions of the reasoning process, rather than simply shortening the entire chain.

### 5.3 CASE STUDY

We present a case study in Figure 6 for qualitative analysis, which shows in detail on the overthinking issue of the original model and the effectiveness of our model mitigating overthinking. From the case we can see that the Qwen3-8B origin model's response involved a total of six reflections in its complete thinking process. During the process, we observe the correct answer was obtained just after two reflections, while the exploration continues and reach the final answer after another four more reflections. This is a clear overthinking example. In contrast, the Qwen3-8B-CER model reflects only once in its thinking process and achieves correct answer even before any reflection. After a brief check of the answer, it directly finalizes the result. Such a process significantly reduces redundant reflection content compared to the origin model. Based on our case observation from Qwen3-4B-CER model, on simple questions, the model may even skip the reflection step and directly determine the final answer, performing rapid reasoning.

## 6  CONCLUSION

This paper addressed the overthinking problem in large language models by introducing a new metric, Token Entropy Cumulative Average (TECA), to measure a model's reasoning exploration. A novel paradigm, "Explore Briefly, Then Decide" was further proposed, which uses a Cumulative Entropy Regulation (CER) mechanism to guide models toward efficient thinking. Experiments on math benchmarks showed our method effectively mitigates overthinking. In future, we plan to explore more sophisticated reward functions and test our approach on a wider range of LLMs with different reasoning mechanisms to further validate its effectiveness.

### ETHICS STATEMENT

This work focuses on improving the efficiency of large language model reasoning by mitigating overthinking through entropy-based regulation. Our study does not involve human subjects, personal data, or sensitive demographic information. The datasets we use (e.g., GSM8K, MATH500, AIME24/25) are publicly available and widely adopted in prior research. We release our code and model checkpoints for reproducibility, ensuring transparency and facilitating future research. We do not anticipate any direct societal harm from our methods; rather, our contributions aim to make reasoning models more efficient and environmentally sustainable by reducing unnecessary computation.

### REPRODUCIBILITY STATEMENT

We provide all necessary details to ensure reproducibility in supplementary materials. The training and evaluation are implemented in the VERL framework with PyTorch. We fine-tuned Qwen3-4B and Qwen3-8B on GSM8K using GRPO with our CER reward design. Code, scripts, and trained model will be released to reproduce our results and figures.

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

## A APPENDIX

### A.1 TRAINING DETAILS

Based on VERL framework, base running script is `run_gsm8k_lora.sh` and the training hyper-parameters are as followed(not mentioned hyper-parameters is default in GRPO algorithm.):

```
1    algorithm.adv_estimator=grpo
2    data.train_batch_size=128
3    data.max_prompt_length=512
4    data.max_response_length=4096
5    data.filter_overlong_prompts=True
6    data.truncation='error'
7    data.shuffle=True
8    actor_rollout_ref.ref.log_prob_use_dynamic_bsz=False
9    actor_rollout_ref.rollout.log_prob_use_dynamic_bsz=False
```

```
10    actor_rollout_ref.model.use_shm=True
11    actor_rollout_ref.model.lora_rank=64
12    actor_rollout_ref.model.lora_alpha=32
13    actor_rollout_ref.actor.optim.lr=3e-6
14    actor_rollout_ref.model.use_remove_padding=True
15    actor_rollout_ref.actor.ppo_mini_batch_size=8
16    actor_rollout_ref.actor.ppo_micro_batch_size_per_gpu=8
17    actor_rollout_ref.actor.use_kl_loss=True
18    actor_rollout_ref.actor.kl_loss_coef=0.001
19    actor_rollout_ref.actor.kl_loss_type=low_var_kl
20    actor_rollout_ref.actor.entropy_coeff=0
21    actor_rollout_ref.model.enable_gradient_checkpointing=True
22    actor_rollout_ref.actor.fsdp_config.param_offload=True
23    actor_rollout_ref.actor.fsdp_config.optimizer_offload=True
24    actor_rollout_ref.rollout.log_prob_micro_batch_size_per_gpu=8
25    actor_rollout_ref.rollout.tensor_model_parallel_size=4
26    actor_rollout_ref.rollout.name=vllm
27    actor_rollout_ref.rollout.temperature=1.5
28    actor_rollout_ref.rollout.top_k=-1
29    actor_rollout_ref.rollout.do_sample=True
30    actor_rollout_ref.rollout.gpu_memory_utilization=0.8
31    actor_rollout_ref.rollout.n=8
32    actor_rollout_ref.rollout.load_format=safetensors
33    actor_rollout_ref.rollout.layered_summon=True
34    actor_rollout_ref.rollout.max_num_batched_tokens=8192
35    actor_rollout_ref.ref.log_prob_micro_batch_size_per_gpu=8
36    actor_rollout_ref.ref.fsdp_config.param_offload=True
37    reward_model.reward_manager="CumulativeEntropyRegulation"
38    algorithm.use_kl_in_reward=False
39    trainer.critic_warmup=0
40    trainer.n_gpus_per_node=4
41    trainer.nnodes=1
42    trainer.save_freq=10
43    trainer.test_freq=0
44    trainer.total_epochs=5
45    trainer.val_before_train=False
```

We conducted our training on 4 H100 GPUs using the VERL reinforcement learning framework and training was performed on the GSM8K training set for about 40 hours. Notably, `reward_model.reward_manager` is a custom reward manager, not using the reward manager provided in the VERL framework.

### A.2 MORE TECA CURVES

In this section, more different TECA curves and evidences and will be presented to support our argument.

#### A.2.1 CORRECT CURVES AND INCORRECT CURVES

In Figure 2, the data used for these curves are all from instances where the model answered correctly. In the main text, we did not show the trend of curves for instances where the model answered incorrectly now showed in Figure 7.

#### A.2.2 EVIDENCE OF EXPLORATION ABILITY AFTER TRAINED

For Llama3.2-3B and Qwen2.5-14B, there is a clear difference in the TECA trend between correctly and incorrectly answered questions. In the "Determination Stage," if these models answer incorrectly, they show an obvious upward trend, meaning the model's uncertainty about the answer increases noticeably. For the overthinking model Qwen3-8B, regardless of whether the answer is correct or not, there is no significant difference in these trends; TECA continues to rise steadily, with incorrect answers showing a more pronounced and uncertain increase. This indicates that for non-overthinking models, there is a certain correlation between whether the model answers correctly and the rise of TECA, while for reasoning models, due to the existence of forking tokens, a rise in TECA is inevitable. However, the problem is that it does not decrease, so we aim to mimic non-overthinking models by ensuring that TECA decreases at the final step. Therefore, CER is designed to suppress TECA at the last step, enabling the model to learn this trend.

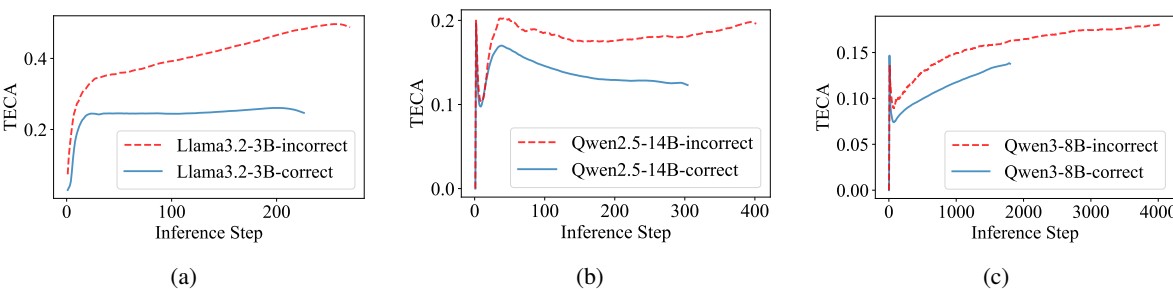

Figure 7: Comparison between correct answers and incorrect answers for Llama3.2-3B, Qwen2.5-14B and Qwen3-8B.

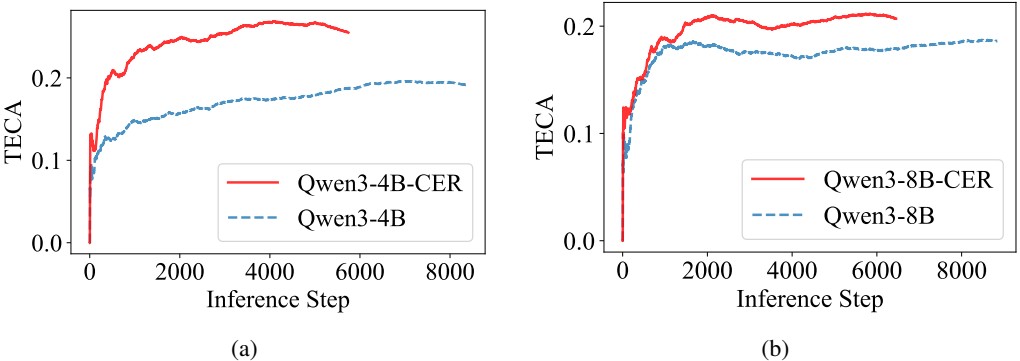

Figure 8: Comparison between origin and trained TECA for Qwen3-4B and Qwen3-8B in AIME dataset.

As showed in Figure 8, after CER, the model's inherent exploration ability is not impaired; on the contrary, it even exhibits a stronger desire for exploration than the original model. The trained model's TECA achieves higher value on the AIME dataset while being significantly shorter in length than the original model. This also demonstrates that the trained model spontaneously reduces a large amount of redundant reasoning and decreases inference depth.

### A.3 DETAILED INFERENCE PROCESS WITH TECA

Here is a sample of the CER model answer and detailed TECA with tokens during inference. We only retain the forking token on x-axis in Figure 9.

<think> Okay, let's see. Janet's ducks lay 16 eggs each day. She uses three of them for breakfast every morning. Then she bakes muffins using four eggs each day. The rest she sells at the market for $2 each. First, calculate total eggs laid per day: 16 eggs. Next, subtract the eggs she uses for breakfast: 16 − 3 = 13 eggs remaining. Then subtract the eggs used for muffins: 13 − 4 = 9 eggs left to sell. Each sold egg brings $2, so total income is 9 2 = $18 per day. </think> Janet's ducks lay 16 eggs per day. She eats 3 eggs for breakfast: 16 − 3 = 13 eggs remain. She uses 4 eggs for muffins: 13 − 4 = 9 eggs remain. She sells these 9 eggs at $2 each: 9 × 2 = 18. #### 18

### A.4 DISMATCHED EXAMPLES

At the level of individual samples, there are some instances where a CER-trained model answers incorrectly on questions it previously solved correctly. The probability of such instance occurring is **only 3.54%**. We conducted case studies on the mismatched instances to investigate the root cause of the errors and found that they primarily stem from the selection of incorrect exploration paths. Since CER does not explicitly regulate the model's exploration trajectory, the adoption of erroneous paths can occur, ultimately leading to incorrect answers. Here are some examples:

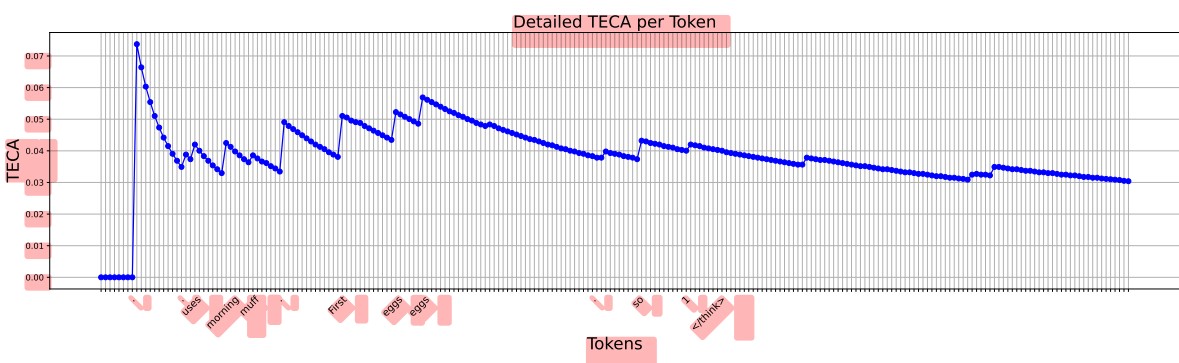

Figure 9: Detailed TECA with tokens. A sample of GSM8K test set on Qwen3-4B-CER.

Example 1:

**Question:** John plans to sell all his toys and use the money to buy video games. He has 13 lego sets and he sells them for $15 each. He ends up buying 8 video games for $20 each and has $5 left. How many lego sets does he still have?

**CER model answer:** `<think>`
n Okay, let's see. John has 13 Lego sets and sells each for $15. So first, I need to calculate how much money he makes from selling the Legos. That would be 13 multiplied by $15. Let me do that math: 13 times 15 is... 195 dollars.Then he uses that money to buy 8 video games, each costing $20. So the total cost for the video games is 8 times 20, which is $160.After buying the games, he has $5 left. So to find out how much money he had initially, I add the cost of the games and the leftover money: $160 plus $5 equals $165.Wait a minute, but he earned $195 from selling the Legos. If he started with $195 and ended up with $165 after buying the games, that makes sense because $195 minus $160 is $35, but he had $5 left, so maybe there's a miscalculation here. Wait no, actually, the total money he has after buying the games is $160 (spent) + $5 (left) = $165. So the initial money from selling Legos was $195, which is more than enough to cover the $160 spent on games and leave $5.But the question asks how many Lego sets he still has. Since he sold all his Legos to buy the games, he should have none left. But wait, maybe he didn't sell all of them? Let me check again.Wait, the problem says he plans to sell all his toys and use the money to buy video games. He has 13 Legos and sells them for $15 each, so he sells all 13, right? Then he buys 8 games with the money and has $5 left. So he should have zero Legos left since he sold all of them. But maybe there's something wrong here?Wait, perhaps I'm missing something. Let me re-express the steps:Total money from selling Legos: $13 * 15 = $195.Money spent on games: $8  20 = $160.Leftover money: $5.So total money used: $160 + $5 = $165. Which is less than $195, meaning he had some money left over, but the question isn't asking about that. The key point is that he sold all his Legos, so he has 0 left.`</think>`John sells all 13 Lego sets for $15 each: $13 \times 15 = 195$ dollars. buys 8 video games for $20 each: $8 \times 20 = 160$ dollars. After purchasing the games, he has $5 left: $195 - 160 \ = \ 35$ dollars initially, but this is unrelated to the number of Lego sets remaining. Since John sold **all** his Lego sets to fund the purchases, he has **none** left. #### $\boxed{0}$

**ground truth:** 2

Example 2:

**Question:** A company pays each of its employees $600 in a month. The company has a policy of increasing the salaries of each of its employees by 10% of the initial salary every year for those who've stayed in the company for five

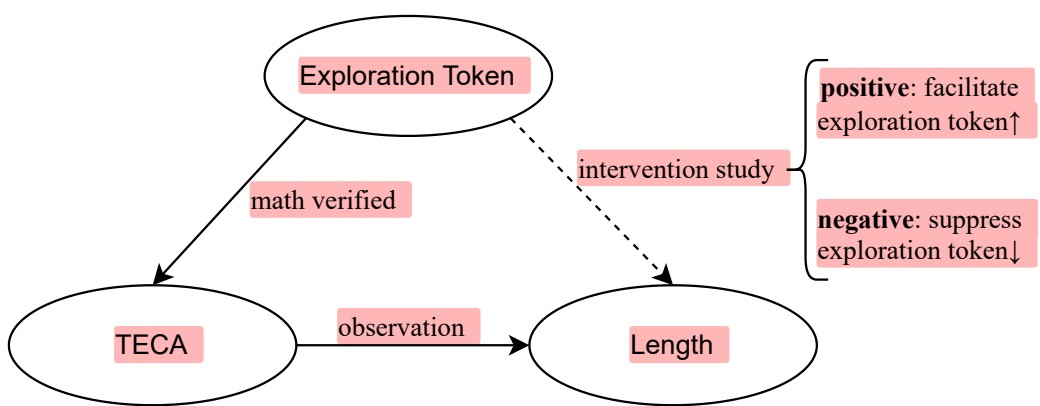

Figure 10: Causal diagnostic framework illustrating the relationships among exploration tokens, TECA, and response length.

years. If Sylvie just clocked 5 years in the company last December, what's her annual salary after three more years of service?

**CER model answer:** <think> Okay, let's see. Sylvie just turned 5 years with the company last December, so she qualifies for the 10% salary increase every year for those who've stayed five years. Her initial monthly salary is $600.First, I need to calculate her annual salary before any increases. Since she earns $600 per month, that's 12 months times $600, which is $7200 per year.Now, since she's been with the company for five years, she gets a 10% increase each year. So for each of those five years, her salary increases by 10% of the initial salary. Wait, does that mean each year her salary is increased by 10% of the original $600 monthly, or 10% of her current salary?The problem says increasing the salaries of each of its employees by 10% of the initial salary every year. So it's 10% of the initial salary each year. So each year, she gets an additional 10% of $600/month.So each year, her monthly salary increases by $600 * 0.10 = 60$. Therefore, after five years, her monthly salary would be $600 + 5$60 = $600 + $300 = $900 per month.Then, her annual salary after five years would be $900 * 12 = 10,800$ per year.Now, the question asks for her annual salary after three more years of service. That means another three years of 10% increases. Each year, again, she gets an additional $60 per month.So after three more years, her monthly salary would be $900 + 3 \times$ $60 = $900 + $180 = $1,080 per month.Therefore, her annual salary would be $1,080 * 12 = 12,960$ per year.</think>Sylvie's initial monthly salary is $600. After 5 years of service, she receives a 10% annual increase based on the initial salary: – **Monthly increase per year**: $600 \times 0.10 = $60$. – **Total monthly salary after 5 years**: $600 + (5 \times 60) = $900$. – **Annual salary after 5 years**: $900 \times 12 = $10,800$. After 3 more years of service, another $60 monthly increase per year applies: – **Total monthly salary after 3 more years**: $900 + (3 \times 60) = $1,080$. – **Final annual salary**: $1,080 \times 12 = $12,960$. ### $\boxed{12960}$

**ground truth:** 9360

## A.5 CAUSAL DIAGNOSTIC

We additionally conduct a causal diagnostic to examine the relationship between TECA trajectories and response length. Figure 10 illustrates the overall framework. Our starting point is the observation that TECA is closely correlated with response length, and that TECA is mathematically derived from token entropy and the occurrence of exploration tokens. From TECA's definition, it is straightforward to verify that the presence of high-entropy exploration tokens contributes directly to increases in TECA due to its cumulative-average formulation. Thus, the remaining

key question is establishing the causal link between exploration tokens and response length. Two prior studies provide complementary intervention-based evidence:

- **Positive intervention**: Wang et al. (2025b) showed that optimizing the RLVR training procedure increases the number of exploration tokens, which in turn leads to longer responses.
- **Negative intervention**: Wang et al. (2025a) demonstrated that suppressing rethinking/exploration tokens at the logit level results in shorter responses.

Taken together, these findings offer convergent causal support for the claim that exploration tokens influence response length. Consequently, they also reinforce the causal relationship between TECA—constructed from token entropy—and response length.

## A.6 AI STATEMENT

We used AI for assisted writing and polishing, including text refinement, translation, text review, and providing revision suggestions. We also utilized AI to search for relevant literature and concepts, and generate reference BibTeX, which were manually verified to ensure they were not fabricated.

