# OpenReview forum: "Explore Briefly, Then Decide: Mitigating LLM Overthinking via Cumulative Entropy Regulation"
_ICLR.cc/2026/Conference — Submitted to ICLR 2026_

### Official Review · Reviewer_nNHM · 2025-10-27

**Soundness:** 4
**Presentation:** 3
**Contribution:** 4
**Rating:** 8
**Confidence:** 4

**Summary:**

The paper observes that in overthinking models, the token entropy cumulative average (TECA) continuously increases during the thinking process. Based on this finding, the authors propose an entropy-based reinforcement learning algorithm that rewards the model based on the TECA during inference, encouraging the model—when producing correct answers—to use as few tokens as possible.

**Strengths:**

1. The paper experimentally demonstrates that TECA exhibits an upward trend during the thinking process of overthinking models.
2. The proposed TECA-based reward mechanism reduces the average output length without significantly compromising model accuracy.

**Weaknesses:**

1. Prior work has shown that models may generate correct answers despite erroneous reasoning steps. Could the proposed method exacerbate this phenomenon?
2. Would reducing TECA negatively impact the model’s exploratory behavior and creativity? Specifically, might the model under-explore and thus underperform on complex problems?
3. The proposed algorithm appears to primarily balance performance and output length relative to the baseline. Could the authors plot results from different models on the same graph to demonstrate that their method indeed lies on the Pareto frontier between accuracy and output length?

**Questions:**

See weakness above.

---

> ### Author Response · Authors · 2025-11-20
> **(1/1)**
>
> Thank you for your valuable suggestions. After carefully considering your comments, we have decided to provide the following corresponding explanations.
>
> > **Weakness 1:** Prior work has shown that models may generate correct answers despite erroneous reasoning steps. Could the proposed method exacerbate this phenomenon?
>
> **Re:** We appreciate the reviewer’s insight regarding the potential **“Right Answer, Wrong Reasoning”** phenomenon, which underscores the delicate balance between reasoning efficiency and reliability. We explore this problem and give the view that the “Right Answer, Wrong Reasoning” phenomenon indicates a behavior of guessing the answer, because the right answer has nothing to do with the wrong reasoning steps.
>
> Theoretically random guessing inherently produces high uncertainty, corresponding to high TECA values. As a result, even if the model obtains the accuracy reward, the TECA reward will remain low if the final-step TECA is not minimized. Therefore, attempting to hack the TECA reward through random guesses is self-defeating, ensuring that the model is incentivized to explore meaningfully before committing to a low-entropy, confident output.
>
> **Evidence from Pass@k:** If the model shows a tendency to guess the answers, the result of pass@k will be different with the pass@1. Because the preference for guessing answers can lead to the loss of logical reasoning of the model. However, pass@k result reveals that Pass@k remains robust despite substantial length reduction, demonstrating that CER preserves the core reasoning capability and will not exacerbate this  “Right Answer, Wrong Reasoning” phenomenon.
>
> |                  | gsm8k/acc | gsm8k/len | aime24/acc | aime24/len |
> | :--------------: | :-------: | :-------: | :--------: | :--------: |
> | 4B-origin-pass@4 |   96.29   |  1373.61  |   73.33    |  11199.52  |
> |  4B-CER-pass@4   |   95.98   |  561.93   |   76.67    |  9325.61   |
> | 4B-origin-pass@8 |   96.89   |  1359.78  |   83.33    |  11328.77  |
> |  4B-CER-pass@8   |   96.36   |  571.70   |   83.33    |  9272.23   |
>
> Overall, based on the experiment results and analysis, we think CER will not exacerbate this phenomenon.
>
> > **Weakness 2:** Would reducing TECA negatively impact the model’s exploratory behavior and creativity? Specifically, might the model under-explore and thus underperform on complex problems?
>
> **Re:** Thank you for your insightful question. We considered this concern at the outset: whether encouraging the model to reduce the final TECA value could inadvertently suppress its exploratory behavior, potentially leading it to underperform on complex problems. Our analysis and experiments indicate that this issue does **not** occur.
>
> Theoretically the segmented reward mechanism is specifically designed to prevent reward hacking, where the model might otherwise output only low-entropy tokens to maximize TECA-related reward. In CER, the model receives the TECA reward **only if it answers correctly**. Consequently, the primary objective remains to **find the correct answer**, which inherently requires exploration. Only after obtaining a correct answer does the model attempt to reduce the final TECA value to maximize reward. This design ensures that CER **encourages necessary exploration** while reducing redundant post-convergence uncertainty, preserving the model’s inherent exploratory capability.
>
> **Experimental evidence:** After CER training, the model’s exploration capability is not weakened. On the contrary, for difficult problems, the CER-trained model exhibits **stronger exploration** than the base model, as illustrated in Figure 8 (A.2.2). For example, in challenging datasets such as AIME, the TECA curve during the reasoning process shows higher values for the CER-trained model, indicating more active exploration. For simpler problems, as shown in Figure 4, the model demonstrates the expected pattern: brief exploration followed by finalizing the answer.
>
> > **Weakness 3:** The proposed algorithm appears to primarily balance performance and output length relative to the baseline. Could the authors plot results from different models on the same graph to demonstrate that their method indeed lies on the Pareto frontier between accuracy and output length?
>
> **Re:** Thank you for this excellent suggestion. We agree that visualizing results on an accuracy-length graph to identify the Pareto Frontier provides the most rigorous demonstration of our method's effectiveness.
>
> In the revised manuscript, we have generated a new figure (Figure 5) that plots Accuracy (Y-axis) against Average Output Length (X-axis) for multiple methods, including baseline approaches and our CER method. The figure clearly shows that the CER-optimized model points lie on the Pareto Frontier, achieving a superior efficiency-performance trade-off compared to the baseline and other standard methods.

---

> > ### Comment · Reviewer_nNHM · 2025-11-27
> > **Thanks**
> >
> > Thank you for your reply and supplementary experiments. The experiments have proved that the method still works well on some more difficult problems. I have no other issues, and I will maintain my score.

---

> > > ### Author Response · Authors · 2025-11-27
> > > **Thanks for your comments**
> > >
> > > We sincerely thank you for your careful review and the constructive feedback provided throughout the process. We truly appreciate the suggestions, which have helped improve the clarity of the paper.

---

### Official Review · Reviewer_oyuY · 2025-10-31

**Soundness:** 2
**Presentation:** 2
**Contribution:** 2
**Rating:** 4
**Confidence:** 4

**Summary:**

This paper introduces a novel framework to address over-thinking in large language models by defining the new metric Token Entropy Cumulative Average (TECA) to quantify exploration during reasoning. Building on TECA, the authors propose the “Explore Briefly, Then Decide” paradigm alongside a Cumulative Entropy Regulation (CER) mechanism, which guides models to terminate reasoning when sufficient certainty is reached. Empirical results on multiple mathematical benchmarks show substantial reductions in response length while maintaining solving capability. The primary contributions are the TECA metric, the CER-based reasoning paradigm, and strong evidence of improved efficiency in reasoning without sacrificing accuracy.

**Strengths:**

1.Introduces a clear, quantitative metric for exploration behavior. TECA provides a novel and interpretable way to monitor when a model is overthinking by measuring uncertainty accumulation during reasoning.
2.Demonstrates large efficiency gains without major accuracy loss. On arithmetic and mathematics benchmarks the method reduced average reasoning length by up to 71 % while maintaining solving performance, showing strong practical value.

**Weaknesses:**

1.The paper focuses heavily on mathematical reasoning benchmarks, but the generalizability of the proposed TECA metric and CER mechanism to open-ended or conversational tasks is not demonstrated.
2.There is limited discussion on the potential trade-off between exploration suppression and creativity or robustness in reasoning. Suppressing exploration might hurt performance on novel or adversarial examples, but this risk is not fully evaluated.
3.In the CER mechanism, the segmentation of reward and the entropy threshold used to determine when to stop exploration or adjust weights appear to require manual tuning. The paper does not include an ablation study examining these hyper-parameters, nor does it describe the detailed workflow for adjusting them.
4.Although response length reduction is compelling, the accuracy drop or change in reasoning style and explanation detail is less thoroughly analyzed, making it harder to assess practical impacts on downstream use-cases.

**Questions:**

1.While CER has generally reduced output length while maintaining accuracy on multiple mathematical benchmarks, does it still suppress meaningful reasoning steps? Does the model exhibit a tendency to halt reasoning prematurely?
2.The current experiments are primarily based on mathematical reasoning benchmarks, but I wonder whether this approach can be generalized to more open-ended or non-mathematical reasoning tasks and undergo broader testing?
3.Based on the paper's description, the Cumulative Entropy Regulation framework appears to involve threshold setting and parameter tuning based on TECA. Could you present a sensitivity analysis demonstrating how different threshold settings impact the results?
4.The authors indicate that while the average response length was significantly reduced across multiple benchmarks, the accuracy rate only experienced a slight decline. However, can accuracy rates alone in mathematical reasoning benchmarks fully represent the completeness and effectiveness of the reasoning process? Does CER potentially lead to the truncation of certain important reasoning exploration steps?
5.The paper only provides theoretical analysis regarding the boundary tokens between exploration and decision-making. Could you present concrete examples of reasoning trajectories that illustrate the transition points between exploration and decision-making? Including such examples in the appendix could better support the theory proposed in the paper.
6.While reasoning efficiency has been improved, does CER training significantly increase training costs? How much additional training cost does it incur compared to the baseline?
7.We note that the experiments were conducted solely on Qwen3-4B and Qwen3-8B models. Can this methodology be extended to larger-scale models or different architectures? Does TECA's effectiveness remain consistent in larger-scale models?
8.I remain skeptical that CER can consistently reduce redundant reasoning while maintaining answer completeness for every test sample. Have there been instances where CER led to performance degradation or premature termination? What mechanisms have the authors implemented to mitigate such issues?
9.The "Explore Briefly, Then Decide" concept proposed in the paper is quite compelling. I wonder whether it could be integrated with other early-exit reasoning mechanisms to achieve better utility?
10.The experimental analysis in the paper appears overly simplistic, lacking ablation studies on components such as the segmented reward mechanism and TECA metric. Furthermore, details regarding the training environment and duration are not provided. Appropriate supplementation in the appendix could substantially improve the completeness of the paper.

---

> ### Author Response · Authors · 2025-11-20
> **(1/6)**
>
> Thank you for your valuable suggestions. After carefully considering your comments, we have decided to provide the following corresponding explanations.
>
> > **Weakness 1**: The paper focuses heavily on mathematical reasoning benchmarks, but the generalizability of the proposed TECA metric and CER mechanism to open-ended or conversational tasks is not demonstrated.
>
> **Re:** Thank you for pointing out this limitation. It is correct that our current experiments focus mainly on mathematical reasoning benchmarks. This choice is motivated by the following considerations:
>
> First, overthinking issues tend to arise in tasks requiring strong multi-step reasoning, which are predominantly mathematical benchmarks. Second, our evaluation follows the setups used in prior work. Existing studies addressing overthinking or aiming to improve LRMs’ reasoning capability/efficiency also primarily evaluate on mathematical datasets [3–9]. To ensure comparability and consistency with these methods, we adopted the same evaluation setup. So the open-ended or conversational tasks are not the main point that such problem focuses on.
>
> However, we appreciate your suggestion regarding broader task types. To examine this, we additionally conducted experiments on a non-mathematical dataset, BoolQ [11], which is a reading comprehension benchmark. The model trained with CER achieved a **45.20% reduction** in output length while maintaining nearly the same accuracy as the base model. This result suggests that CER can extend beyond mathematical reasoning settings and remains effective on textual tasks.
>
> | model        | accuracy | length |
> | ------------ | :------: | :----: |
> | qwen3-4B     |  87.40   | 403.43 |
> | qwen3-4B-CER |  87.19   | 221.08 |
>
> > **Weakness 2:** There is limited discussion on the potential trade-off between exploration suppression and creativity or robustness in reasoning. Suppressing exploration might hurt performance on novel or adversarial examples, but this risk is not fully evaluated.
>
> **Re:** We thank the reviewer for this observation. By suppressing the final-step TECA value, CER reduces the model’s tendency to over-explore. We carefully considered whether this could compromise the model’s inherent exploratory capability or creativity, and our analysis shows that it does not.
>
> **Theoretical perspective:**
> The segmented reward mechanism is explicitly designed to preserve exploration while limiting overthinking. The model receives the TECA-related reward **only after producing a correct answer**. As a result, the model’s primary objective remains to solve the problem correctly, which inherently requires exploration—especially in Long CoT models, where multi-step reasoning is essential. Once the correct answer is obtained, CER then encourages reducing the final TECA value. This process teaches the model to explore sufficiently to find the correct answer, and only then minimize unnecessary late-stage exploration. Therefore, CER theoretically does not impair the model’s inherent ability to explore.
>
> **Experimental perspective:**
> Experimental results confirm that CER-trained models retain, and in some cases enhance, exploration on challenging problems. For example, Figure 8 (Appendix A.2.2) shows that on difficult datasets such as AIME, CER-trained models exhibit higher TECA values during the early reasoning stages compared to the base model, indicating stronger exploratory behavior. On simpler problems, as illustrated in Figure 4, the TECA curve exhibits the expected pattern: brief exploration followed by answer finalization.
>
> **Conclusion:**
> CER does not suppress the model’s fundamental exploratory capability. Instead, it encourages **moderate exploration**, reducing over-exploration while preserving and even enhancing effective exploration when needed.

---

> ### Author Response · Authors · 2025-11-20
> **(2/6)**
>
> > **Weakness 3:** In the CER mechanism, the segmentation of reward and the entropy threshold used to determine when to stop exploration or adjust weights appear to require manual tuning. The paper does not include an ablation study examining these hyper-parameters, nor does it describe the detailed workflow for adjusting them.
>
> **Re:** Thank you for the question. There appears to be a misunderstanding regarding the implementation of CER.
>
> CER consists of two main components:
>
> 1. **Rewards:** CER uses two types of reinforcement learning rewards: the **accuracy reward** and the **TECA reward**. The accuracy reward is 1 if the answer is correct and 0 otherwise. The TECA reward is inversely proportional to the final-step TECA value: a lower final-step TECA corresponds to a higher reward.
> 2. **Segmented Reward Mechanism (SRM):** The model only receives the TECA reward if the answer is correct. If the answer is incorrect, the TECA reward is withheld, even if the final-step TECA value is low.
>
> Importantly, CER does **not** use any entropy threshold or manually instruct the model when to stop exploration. The degree of exploration is entirely controlled by the model itself. CER simply encourages the model to reach the correct answer and be as confident as possible, without interfering with its internal exploration process.
>
> > **Weakness 4:** Although response length reduction is compelling, the accuracy drop or change in reasoning style and explanation detail is less thoroughly analyzed, making it harder to assess practical impacts on downstream use-cases
>
> **Re:** Thank you for the comment. The fluctuations in accuracy observed after CER training are related to the nature of our training strategy. As explained in the response to Weakness 3, CER encourages the model to be as confident as possible in its answer, **provided the answer is correct**.
>
> As a result, the model develops a reasoning style in which it first engages in exploration to solve the problem, then consolidates a final answer after reflection, and subsequently maintains confidence without further reconsideration. This self-determined process can occasionally lead to **misjudgments**, where the model believes its answer is correct but it is actually incorrect. These misjudgments explain the observed fluctuations in accuracy.
>
> Importantly, this behavior is consistent with natural reasoning: even humans can occasionally commit to incorrect answers after reflection. Therefore, the minor accuracy fluctuations reflect the model’s independent reasoning process rather than a degradation in overall performance.

---

> ### Author Response · Authors · 2025-11-20
> **(3/6)**
>
> > **Question 1:** While CER has generally reduced output length while maintaining accuracy on multiple mathematical benchmarks, does it still suppress meaningful reasoning steps? Does the model exhibit a tendency to halt reasoning prematurely?
>
> **Re:** Thank you for the question.
>
> **1. CER suppresses unnecessary reasoning, not meaningful steps:**
> CER is designed to compress **redundant reasoning steps** in the model’s inference process. For simple problems, once the model has obtained the correct answer, additional reflection steps are unnecessary. These excessive steps constitute what we define as **meaningless reasoning**. For difficult problems, plenty of reasoning steps is necessary, so the improvement of length is limited and the performance keeps nearly the same. CER reduces redundant steps while preserving essential reasoning.
>
> For example, as shown in Figure 6, before training, the model performed four additional reflections after obtaining the correct answer. After CER training, it conducted only one brief reflection to confirm the answer, eliminating redundant checks without impacting correctness.
>
> Further evidence comes from **Accuracy and Pass@k metrics**:
>
> - If CER indiscriminately truncated necessary reasoning, we would expect a significant drop in accuracy. However, our results show up to **71% reduction in response length** while maintaining solving capability, indicating that eliminated tokens were indeed redundant.
> - **Pass@k results** confirm that CER reduces length without sacrificing the breadth of sampling, demonstrating that CER suppresses useless branches rather than prematurely cutting off meaningful exploration.
>
>
> |                  | gsm8k/acc | gsm8k/len | aime24/acc | aime24/len |
> | :--------------: | :-------: | :-------: | :--------: | :--------: |
> | 4B-origin-pass@4 |   96.29   |  1373.61  |   73.33    |  11199.52  |
> |  4B-CER-pass@4   |   95.98   |  561.93   |   76.67    |  9325.61   |
> | 4B-origin-pass@8 |   96.89   |  1359.78  |   83.33    |  11328.77  |
> |  4B-CER-pass@8   |   96.36   |  571.70   |   83.33    |  9272.23   |
>
> **2. CER does not induce premature stopping:**
> The CER-trained model does not halt reasoning early. Figure 4 and Figure 8 (Appendix A.2.2) show that:
>
> - On **simple problems**, the model stops once it is confident in its answer, avoiding overthinking.
> - On **difficult problems**, the TECA curve remains high during exploration, reflecting continued search for the correct answer. Correspondingly, response length is not excessively reduced.
>
> This behavior is consistent with the design of CER: **the model prioritizes obtaining the correct answer, and only then focuses on increasing confidence. When the correct answer has not yet been reached, the model continues its reasoning rather than exiting prematurely, thereby preserving all essential exploration.**
>
> > **Question 2:** The current experiments are primarily based on mathematical reasoning benchmarks, but I wonder whether this approach can be generalized to more open-ended or non-mathematical reasoning tasks and undergo broader testing?
>
> **Re:** We provide detailed clarification to the question in our response to **Weakness 1**.

---

> ### Author Response · Authors · 2025-11-20
> **(4/6)**
>
> > **Question 3:** Based on the paper's description, the Cumulative Entropy Regulation framework appears to involve threshold setting and parameter tuning based on TECA. Could you present a sensitivity analysis demonstrating how different threshold settings impact the results?
>
> **Re:** Thank you for the question. As clarified in our response to **Weakness 3**, CER does **not** rely on manually set entropy thresholds; the degree of exploration is determined by the model itself.
>
> To further address sensitivity, we conducted additional ablation experiments focusing on **module contributions** and **temperature settings during rollout**.
>
> **1. Module ablation:**
> CER consists of two components: the **TECA reward** and the **Segmented Reward Mechanism (SRM)**, with SRM depending on TECA. We evaluated three configurations:
>
> - **Full CER (TECA + SRM)**
> - **CER without SRM**
> - **Accuracy-only reward (no TECA)**
>
> The results are as follows:
>
> |                     | GSM8K/acc | GSM8K/len | AIME24/acc | AIME24/len |
> | :-----------------: | :-------: | :-------: | :--------: | :--------: |
> |    qwen3-4B-CER     |   94.09   |  391.08   |   61.11    |  9215.77   |
> | qwen3-4B-remove-SRM |   91.89   |  542.61   |   62.22    |  8750.14   |
> |  qwen3-4B-acc-only  |   95.22   |  887.07   |   64.44    |  10953.92  |
>
> We observed that: (1) Removing both TECA and SRM increases response length despite slightly higher accuracy. (2) Removing SRM alone reduces length partially but decreases accuracy on simple questions, with minor improvement on difficult questions. (3) Slight fluctuations on difficult questions are consistent with the reasoning dynamics described in **Weakness 4**.
>
> **2. Temperature sensitivity during rollout:**
>
> We also analyzed the effect of temperature, which influences exploration step size during training: higher temperature induces more diverse exploration and aggressive parameter updates. Results are summarized below:
>
> | temperature | GSM8K/acc | GSM8K/len | AIME24/acc | AIME24/len |
> | ----------- | --------- | --------- | ---------- | ---------- |
> | 1.5         | 94.09     | 391.08    | 61.11      | 9215.77    |
> | 1.2         | 93.71     | 456.44    | 61.11      | 8703.79    |
> | 0.8         | 93.71     | 606.27    | 66.67      | 9646.43    |
>
> We observed that: (1) For simple questions, higher temperature improves both length reduction and accuracy slightly. (2) For difficult questions, temperature changes have limited impact on length, as substantial reflection is still necessary to solve challenging problems.
>
> > **Question 4:** The authors indicate that while the average response length was significantly reduced across multiple benchmarks, the accuracy rate only experienced a slight decline. However, can accuracy rates alone in mathematical reasoning benchmarks fully represent the completeness and effectiveness of the reasoning process? Does CER potentially lead to the truncation of certain important reasoning exploration steps?
>
> **Re:** We thank the reviewer for this important question, which addresses the trade-off between reasoning reliability and efficiency. We agree that final-answer accuracy alone does not fully capture the completeness and effectiveness of the reasoning process, especially in Chain-of-Thought (CoT) generation. CER is explicitly designed to reduce **redundant exploration**, not essential reasoning steps, and this is supported by multiple lines of evidence:
>
> **1. Pass@k Results:**
> If CER were prematurely truncating important reasoning, Pass@k—reflecting the model’s ability to find correct answers across multiple samples—would degrade. Our results show that Pass@k remains robust despite substantial length reduction, demonstrating that CER preserves the core reasoning capability while eliminating unnecessary detours and post-convergence reflections.
>
> |                  | gsm8k/acc | gsm8k/len | aime24/acc | aime24/len |
> | :--------------: | :-------: | :-------: | :--------: | :--------: |
> | 4B-origin-pass@4 |   96.29   |  1373.61  |   73.33    |  11199.52  |
> |  4B-CER-pass@4   |   95.98   |  561.93   |   76.67    |  9325.61   |
> | 4B-origin-pass@8 |   96.89   |  1359.78  |   83.33    |  11328.77  |
> |  4B-CER-pass@8   |   96.36   |  571.70   |   83.33    |  9272.23   |
>
> **2. Pareto Frontier Analysis:**
> The Accuracy-Length Pareto Frontier (Figure 5 in latest version) confirms that CER achieves a superior efficiency-performance trade-off. The CER-optimized model points lie on the frontier, showing that the retained reasoning steps are highly effective and represent a minimal set required for high performance.
>
> **3. Mechanistic Rationale via TECA:**
> TECA’s cumulative averaging ensures that **necessary exploratory steps** (high initial entropy) are tolerated. CER primarily penalizes trajectories where TECA remains unnecessarily high **after the exploration-to-determination transition**, effectively pruning only the redundant tail tokens while preserving essential problem-solving steps.

---

> ### Author Response · Authors · 2025-11-20
> **(5/6)**
>
> > **Question 5:** The paper only provides theoretical analysis regarding the boundary tokens between exploration and decision-making. Could you present concrete examples of reasoning trajectories that illustrate the transition points between exploration and decision-making? Including such examples in the appendix could better support the theory proposed in the paper.
>
> **Re:** Thank you for the constructive suggestion. We agree that providing concrete examples of reasoning trajectories can more directly illustrate the transition from exploration to decision-making.
>
> While the main text includes case studies highlighting changes in response length and reasoning, these examples primarily convey text-level differences. To better visualize the process, we propose combining **real-time TECA curves** with the model’s generated tokens, allowing an intuitive view of the transition from exploration to decision-making.
>
> Specifically, we will plot extended TECA curves with the **horizontal axis representing inference steps** and the **vertical axis representing TECA values**. Key tokens corresponding to each step will be annotated along the horizontal axis to clearly mark the exploration-to-decision transition. An example of this visualization is shown in Figure 9 of the revised version.
>
> > **Question 6:** While reasoning efficiency has been improved, does CER training significantly increase training costs? How much additional training cost does it incur compared to the baseline?
>
> **Re:** We thank the reviewer for this important question. CER training does increase training overhead, primarily in terms of **GPU memory usage**.
>
> We use **verl** as our training framework. To implement CER, we modified the data flow in verl to retain and propagate token entropy during training, which is essential for calculating the TECA reward. In the original training process, computed entropy is typically discarded immediately after calculating `log_probs`. In contrast, CER requires this entropy for subsequent reward computation, preventing its deletion.
>
> As a result, **from the stage of log_probs computation to reward calculation**, memory usage is significantly higher compared to the baseline. This explains the additional training cost incurred by CER.
>
> > **Question 7:** We note that the experiments were conducted solely on Qwen3-4B and Qwen3-8B models. Can this methodology be extended to larger-scale models or different architectures? Does TECA's effectiveness remain consistent in larger-scale models?
>
> **Re:** We thank the reviewer for raising this important question regarding the **generalizability and scalability** of CER. We agree that evaluating performance across a broader range of models is crucial.
>
> **Model selection rationale:** CER is specifically designed to address the **overthinking problem**, which primarily occurs in Long CoT models. Therefore, we focused on Long CoT models, mainly choosing **Llama series** and the **Qwen series**. Most Llama models are not Long CoT models and do not exhibit overthinking, so we selected Qwen models for our experiments. Besides, we follow such model setting by others' work[5-10].
>
> We also attempted to apply CER to other models, such as **R1-distill-Qwen2.5-7B** and **Llama3.2-3B**. As expected, post-training results showed minimal length reduction and even lead to serious length problem because of the absence of exploration ability:
>
> |                           | gsm8k/acc | gsm8k/len | aime24/acc | aime24/len |
> | ------------------------- | --------- | --------- | ---------- | ---------- |
> | R1-distill-Qwen2.5-7B     | 56.48     | 525.44    | 40         | 8852.80    |
> | R1-distill-Qwen2.5-7B-CER | 73.01     | 459.11    | 33.33      | 8797.03    |
> | Llama3.2-3B               | 17.51     | 215.78    | 3.33       | 935.30     |
> | Llama3.2-3B-CER           | 75.82     | 599.23    | 3.33       | 10805.37   |
>
> **Scalability to larger models:**
> Due to computational constraints, our experiments focused on Qwen3-4B and 8B models. Nevertheless, the **theoretical foundation** of CER suggests broad applicability. CER relies on the existence of **forking tokens** during exploration [8]. As long as a model can perform exploratory reasoning, CER can be applied to mitigate overthinking. Moreover, CER is model-agnostic:
>
> - **TECA Mechanism:** TECA is computed from token logits, enabling Shannon Entropy calculation. Since virtually all Transformer-based LLMs output logits, TECA can be applied across model sizes and architectures.
> - **CER Integration:** CER is implemented as a reinforcement learning reward signal within the GRPO framework. Any model supporting standard policy gradient optimization can integrate CER to shape its reasoning policy.
>
> **Future validation:** We acknowledge the importance of generalizability and scalability. We are also working on optimizing the verl-based implementation to reduce memory usage, enabling broader experimentation in the future.

---

> ### Author Response · Authors · 2025-11-20
> **(6/6)**
>
> > **Question 8:** I remain skeptical that CER can consistently reduce redundant reasoning while maintaining answer completeness for every test sample. Have there been instances where CER led to performance degradation or premature termination? What mechanisms have the authors implemented to mitigate such issues?
>
> **Re:** Thank you for your insightful question. On average, CER substantially reduces redundant reasoning steps **without significant performance degradation**. However, at the level of individual samples, there are rare instances where a CER-trained model answers incorrectly on questions it previously solved correctly. The probability of such exceptions is very low.
>
> We calculated the occurrence frequency of such mismatch on GSM8K test set with repeated sampling for robustness and got that mismatch probability is **only 3.54%**. Mismatch probability's formula is $\text{Mismatch Probability} = \frac{\text{mismatched sample number}}{\text{all number}}$.
>
> We also conducted case studies on the mismatched instances to investigate the root cause of the errors and found that they primarily stem from the selection of incorrect exploration paths. Since CER does not explicitly regulate the model's exploration trajectory, the adoption of erroneous paths can occur, ultimately leading to incorrect answers. As discussed in our response to Weakness 4, this behavior is within expectations; therefore, we have not implemented specific mechanisms to mitigate this limitation in the current scope. However, we plan to investigate the underlying causes of such path deviations and explore mechanisms to regulate them in future work. Detailed examples of these error cases and corresponding analyses are provided in Appendix A.4 of the revised manuscript
>
> > **Question 9:** The "Explore Briefly, Then Decide" concept proposed in the paper is quite compelling. I wonder whether it could be integrated with other early-exit reasoning mechanisms to achieve better utility?
>
> **Re:** Thank you for your constructive suggestion. Many early-exit methods require determining when to exit, and TECA can serve as an auxiliary signal for this judgment, such as DEER[10].
>
> Furthermore, we conducted additional experiments integrating CER with other overthinking-mitigation mechanisms such as CCoT and CoD. The results show that combining CER with these approaches yields a **synergistic reduction in response length**, confirming CER’s effectiveness as a length-control component. However, the **accuracy improvements are not consistently robust** across tasks:
>
> |          | gsm8k/acc | gsm8k/len | aime24/acc | aime24/len |
> | -------- | :-------: | :-------: | :--------: | :--------: |
> | CCoT     |   82.56   |  616.42   |   56.67    |  9491.87   |
> | CER+CCoT |   90.14   |  299.61   |   50.00    |  5870.47   |
> | CoD      |   93.30   |  385.50   |   23.30    |  3607.83   |
> | CER+CoD  |   90.60   |  179.76   |   26.67    |  2637.97   |
>
>  For future work, we plan to investigate **more principled integration strategies** that preserve CER’s length-reduction benefits while maintaining stronger accuracy robustness.
>
> > **Question 10:** The experimental analysis in the paper appears overly simplistic, lacking ablation studies on components such as the segmented reward mechanism and TECA metric. Furthermore, details regarding the training environment and duration are not provided. Appropriate supplementation in the appendix could substantially improve the completeness of the paper.
>
> **Re:** Thank you for your constructive feedback. For ablation studies, we have provided specific presentation and explanation in our response to **Question 3**. For the details of the training environment and duration, we conducted our training on 4 H100 GPUs using the verl reinforcement learning framework. The hyperparameter settings are as described in Appendix A.1, and training was performed on the GSM8K training set for ~40 hours. We will add this information to the appendix in the subsequent revised version.
>
> &ensp;
>
> **Reference:**
>
> [1] Do NOT Think That Much for 2+3=? On the Overthinking of o1-Like LLMs
>
> [2] Stop Overthinking: A Survey on Efficient Reasoning for Large Language Models
>
> [3] Sample More to Think Less:  Group Filtered Policy Optimization for Concise Reasoning
>
> [4] Wait, We Don’t Need to “Wait”!  Removing Thinking Tokens Improves Reasoning Efficiency
>
> [5] When More is Less: Understanding Chain-of-Thought Length  in LLMs
>
> [6] Efficient Reasoning for Large Reasoning Language Models via Certainty-Guided Reflection Suppression
>
> [7] Pruning Long Chain-of-Thought of Large Reasoning Models via Small-Scale Preference Optimization
>
> [8] Beyond the 80/20 Rule: High-Entropy Minority Tokens Drive Effective Reinforcement Learning for LLM Reasoning
>
> [9] The Entropy Mechanism of Reinforcement Learning for Reasoning Language Models
>
> [10] Dynamic Early Exit in Reasoning Models
>
> [11]  BoolQ: Exploring the Surprising Difficulty of Natural Yes/No Questions

---

### Official Review · Reviewer_6psw · 2025-11-01

**Soundness:** 2
**Presentation:** 2
**Contribution:** 2
**Rating:** 2
**Confidence:** 4

**Summary:**

This paper proposes TECA as a proxy for “exploration” during CoT inference, and introduces a GRPO-based reward shaping scheme (CER) that suppresses high TECA at the tail of the trajectory. Experiments show substantial reductions in average CoT length with minimal accuracy degradation.

**Strengths:**

The TECA metric is intuitively interpretable, simple to compute, and shows reasonable empirical correlation with response length.

The qualitative case study indeed shows fewer redundant “reflective” tokens in the CER-trained model.

**Weaknesses:**

**W1. Experimental scale is insufficient to support the claimed generality.**
Only Qwen3-4B and Qwen3-8B are used. GRPO RL training is notoriously sensitive to backbone initialization and emergent base-model bias. A two-model sample is not enough to justify the claim that the proposed mechanism reflects a general property of “overthinking” and adaptive reasoning. In fact, even for the 8B model, CER only provides a marginal advantage relative to CCoT in some metrics.

**W2. The work should also report Pass@k.**
Length reduction could come from eliminating alternative chains prematurely, not from eliminating irrelevant exploration. Pass@k is the standard instrument to distinguish “more decisive reasoning” from “shallower sampling.” Without this control, it is unclear whether CER suppresses useless branches or simply cuts off exploration globally.

**W3. Conceptual analysis is shallow.**
Preliminaries section repeats known GRPO and entropy definitions, while the core insight, *TECA is a principled proxy of exploration*, is not justified beyond curve visualizations. Two major omissions weaken the conceptual claim:
- The paper only establishes correlation between TECA trajectories and response length. There is no causal diagnostic: no intervention study (injecting entropy spikes), no ablation vs. simpler uncertainty surrogates (e.g. logit margin / variance). Thus, it is not shown that TECA is the right variable. It may simply correlate with the visible surface symptom (redundant tokens).
- The CER reward construction is not theoretically motivated. Rewarding TECA only on correct samples can encourage lucky short guesses, and the paper does not analyze this incentive misalignment. The design choices (equal weighting with accuracy reward, exponential form, applying TECA only at the end step) are hyperparameter-like knobs but are not justified mathematically.

**Questions:**

see weakness

---

> ### Author Response · Authors · 2025-11-20
> **(1/3)**
>
> Thank you for your valuable comments. After carefully considering your suggestions, we have decided to provide the following corresponding explanation.
>
> > **Weakness 1:** Experimental scale is insufficient to support the claimed generality. Only Qwen3-4B and Qwen3-8B are used. GRPO RL training is notoriously sensitive to backbone initialization and emergent base-model bias. A two-model sample is not enough to justify the claim that the proposed mechanism reflects a general property of “overthinking” and adaptive reasoning. In fact, even for the 8B model, CER only provides a marginal advantage relative to CCoT in some metrics.
>
> **Re:** We thank the reviewer for raising this important point. The concern focuses on whether CER can generalize beyond Qwen3-4B and Qwen3-8B, given the sensitivity of GRPO RL training to backbone initialization and base-model biases.
>
> First, it is important to clarify that CER specifically addresses the overthinking problem, which primarily arises in Long CoT models. CER also requires access to model output logits during training, which limits its application to open-source Long CoT models. We selected two mainstream families—Qwen and Llama—for evaluation. Within the Qwen series, models with Long CoT capability include Qwen3 and QWQ; for timeliness, we focused on Qwen3. The Llama series, as shown in Figure 2, does not exhibit overthinking due to the absence of Long CoT behavior, so applying CER to these models is not meaningful in practice.
>
> We also explored applying CER to other models, such as R1-distill-Qwen2.5-7B and Llama3.2-3B. For R1-distill-Qwen2.5-7B, we found limited improvement in the response length because it doesn't have severe overthinking problem. Due to the absence of exploration ability, applying CER to Llama3.2-3B doesn't work. Even it leads to a serious problem as showed in the table: on aime24 dataset, the response length is 10 times longer without any accuracy improvement.
>
> |                           | gsm8k/acc | gsm8k/len | aime24/acc | aime24/len |
> | ------------------------- | --------- | --------- | ---------- | ---------- |
> | R1-distill-Qwen2.5-7B     | 56.48     | 525.44    | 40         | 8852.80    |
> | R1-distill-Qwen2.5-7B-CER | 73.01     | 459.11    | 33.33      | 8797.03    |
> | Llama3.2-3B               | 17.51     | 215.78    | 3.33       | 935.30     |
> | Llama3.2-3B-CER           | 75.82     | 599.23    | 3.33       | 10805.37   |
>
> [1] undertook a pioneering exploration of RLVR through the novel perspective of token entropy patterns, comprehensively analyzing how different tokens influence reasoning performance. They found that only a small fraction of tokens exhibit high entropy, and these tokens(i.e. forking tokens) act as critical forks that steer the model toward diverse reasoning pathways.
>
> Based on the discovery, CER’s only prerequisite is that the model supports exploration(i.e. existence of forking token). Under this condition, the method can effectively address overthinking. Importantly, CER is **model-agnostic** in its design:
>
> - **TECA Mechanism:** TECA computes Shannon Entropy directly from token logits. Since virtually all modern Transformer-based LLMs output logits, TECA can be universally applied regardless of model architecture or size.
> - **CER Integration:** CER is implemented as a reinforcement learning reward within the GRPO framework. Any model supporting standard policy-gradient optimization can adopt CER to shape its reasoning policy.
>
> > **Weakness 2:** The work should also report Pass@k. Length reduction could come from eliminating alternative chains prematurely, not from eliminating irrelevant exploration. Pass@k is the standard instrument to distinguish “more decisive reasoning” from “shallower sampling.” Without this control, it is unclear whether CER suppresses useless branches or simply cuts off exploration globally.
>
> **Re:** Thank you for this insightful comment. We have supplemented our experiments with Pass@k evaluation and provide the following analysis.
>
> If CER truncated important exploration, the overall ability to find the correct answer across multiple samples (Pass@k) should drop significantly. Our new results show that **Pass@k remains robust**, despite the substantial length reduction. This demonstrates that CER successfully retains the model's core ability to find the correct reasoning path while eliminating unnecessary detours and post-convergence chatter.
>
> |                  | gsm8k/acc | gsm8k/len | aime24/acc | aime24/len |
> | :--------------: | :-------: | :-------: | :--------: | :--------: |
> | 4B-origin-pass@4 |   96.29   |  1373.61  |   73.33    |  11199.52  |
> |  4B-CER-pass@4   |   95.98   |  561.93   |   76.67    |  9325.61   |
> | 4B-origin-pass@8 |   96.89   |  1359.78  |   83.33    |  11328.77  |
> |  4B-CER-pass@8   |   96.36   |  571.70   |   83.33    |  9272.23   |

---

> ### Author Response · Authors · 2025-11-20
> **(2/3)**
>
> > Conceptual analysis is shallow. Preliminaries section repeats known GRPO and entropy definitions, while the core insight, *TECA is a principled proxy of exploration*, is not justified beyond curve visualizations. Two major omissions weaken the conceptual claim:
> >
> > **Weakness 3.1:** The paper only establishes correlation between TECA trajectories and response length. There is no causal diagnostic: no intervention study (injecting entropy spikes), no ablation vs. simpler uncertainty surrogates (e.g. logit margin / variance). Thus, it is not shown that TECA is the right variable. It may simply correlate with the visible surface symptom (redundant tokens).
>
> **Re:** Thank you for this insightful comment. We address the two subpoints below.
>
> **1. Causal diagnostic and Intervention study:** Thanks for the suggestion. Our current results show that TECA rises when high-entropy exploration tokens (i.e., forking/rethinking tokens) appear, and because TECA is a cumulative average, even a small number of such spikes will noticeably increase the curve. To strengthen the causal interpretation, we refer to two complementary lines of evidence from prior work as the intervention study:
>
> - **Positive intervention:** RLVR optimization in [1] **increases the prevalence of forking tokens**. This intervention simultaneously **leads to longer responses**, despite improved task performance. This indicates that encouraging exploration tokens causally pushes models toward producing lengthier outputs, consistent with the TECA trajectory we observe.
> - **Negative intervention:** Conversely, [5] explicitly **suppresses rethinking/forking tokens at the logit level** and successfully **reduces response length**. This provides the opposite directional causal evidence: reducing exploration spikes leads to shorter outputs.
>
> Together, these studies provide converging causal support for the connection between exploration tokens and response length which is also correlation between TECA and response length. Detailed analysis can be found in A.5 in latest manuscript.
>
> **2. Comparison with simpler uncertainty surrogates (logit margin/variance):**
> TECA has a stronger theoretical foundation than logit margin or variance. Logit margin only considers the top two logits, ignoring other informative logits, and thus cannot capture the full distributional uncertainty, especially in the presence of forking tokens. Logit variance considers all logits but can misrepresent uncertainty, as shown below:
>
> - Consider an unnormalized logit vector of length 5 with values in `[0,10]`:
>   - **Logit variance:**
>     - `[5,5,5,5,5]` → min variance = 0 (most uncertain)
>     - `[0,0,10,10,10]` → max variance = 24
>     - `[10,0,0,0,0]` → variance = 16 (most certain)
>   - **Token entropy:**
>     - `[10,0,0,0,0]` → min value (most certain)
>     - `[5,5,5,5,5]` → max value (most uncertain)
>
> This demonstrates that logit variance fails to consistently represent extreme uncertainty scenarios, making it unreliable as a training signal.
>
> TECA, by contrast, is the cumulative sum of step-wise token entropy. It captures **global uncertainty** across the reasoning process, allowing early-stage exploration while penalizing excessive late-stage exploration. In contrast, logit margin or variance only provide local information; using them would force the model to minimize entropy indiscriminately, potentially eliminating necessary exploration.

---

> ### Author Response · Authors · 2025-11-20
> **(3/3)**
>
> > Conceptual analysis is shallow. Preliminaries section repeats known GRPO and entropy definitions, while the core insight, *TECA is a principled proxy of exploration*, is not justified beyond curve visualizations. Two major omissions weaken the conceptual claim:
> >
> > **Weakness 3.2:** The CER reward construction is not theoretically motivated. Rewarding TECA only on correct samples can encourage lucky short guesses, and the paper does not analyze this incentive misalignment. The design choices (equal weighting with accuracy reward, exponential form, applying TECA only at the end step) are hyperparameter-like knobs but are not justified mathematically.
>
> **Re:** Thank you for raising this question. We provide a detailed response below.
>
> **1. Theoretical motivation of CER reward construction:**
> The theoretical foundation of TECA has been discussed in response to Weakness 3.1. Building on this, CER’s reward mechanism is designed to guide the model’s exploration behavior through two key components:
>
> - **TECA reward:** TECA captures the model’s exploration process: an increase reflects ongoing exploration, while a decrease indicates confirmation of an answer. As shown in Figure 2, overthinking models exhibit consistently rising TECA curves, whereas non-overthinking models tend to have a downward or stable trend at the end. The TECA reward is therefore designed to minimize the final TECA value, encouraging overthinking models to converge and confirm their answer in the final stage. To ensure the reward scale aligns with standard RL practice and with the accuracy reward range [0,1], we apply a base-*e* exponential function to the negative final TECA value, i.e., $e^{-\text{TECA}_{-1}}$, constraining the reward within a comparable range.
> - **SRM:** SRM ensures that the model retains exploration capability while learning to conclude efficiently. The model must first produce a correct answer to receive the TECA reward. During this process, it engages in exploration as needed. Once the correct answer is reached, the TECA reward encourages the model to output low-entropy tokens, reducing unnecessary late-stage exploration and shortening response length.
>
> **2. Rewarding TECA only on correct samples and potential for lucky guesses:**
> It is true that rewarding TECA only on correct samples might seem to incentivize guessing. However, this does not undermine the design. Random guessing inherently produces high uncertainty, corresponding to high TECA values. As a result, even if the model obtains the accuracy reward, the TECA reward will remain low if the final-step TECA is not minimized. Therefore, attempting to hack the TECA reward through random guesses is self-defeating, ensuring that the model is incentivized to explore meaningfully before committing to a low-entropy, confident output.
>
> &nbsp;
>
> **Reference:**
>
> [1] Beyond the 80/20 Rule: High-Entropy Minority Tokens Drive Effective Reinforcement Learning for LLM Reasoning
>
> [2] The Unreasonable Effectiveness of Entropy Minimization in LLM Reasoning
>
> [3] One-shot Entropy Minimization
>
> [4] A mathematical theory of communication
>
> [5] Wait, We Don’t Need to “Wait”!  Removing Thinking Tokens Improves Reasoning Efficiency

---

> > ### Comment · Reviewer_6psw · 2025-11-27
> >
> > I think this manuscript feels rushed, and the rebuttal does not adequately resolve the core concerns. The following points summarize the remaining issues.
> >
> > **W1: Insufficient evidence for generality**
> >
> > It is difficult to accept the unsubstantiated claim:
> >
> > > “For R1-distill-Qwen2.5-7B, we found limited improvement in the response length because it doesn’t have a severe overthinking problem.”
> >
> > The authors provided no prior literature or reported experiment supporting this statement. If only the two models tested in the paper exhibit the “overthinking” phenomenon, then the motivation of studying this phenomenon becomes unclear. Overthinking is indeed an important behavioral pattern, but before proposing a mitigation method, one must rigorously establish which models exhibit it.
> >
> > ---
> >
> > **W2: Pass@k evaluation remains unclear and incomplete**
> >
> > You need to clarify how length is aggregated across k samples in the Pass@k evaluation in the table (e.g., mean or minimum among k, etc.).
> >
> > In addition, Pass@8 is far too small to draw meaningful conclusions. As shown in He et al. (2025) [1], Pass@k becomes informative when k ≥ 32, especially for math reasoning tasks. I fully understand that running large k sampling on models is expensive, but AIME24 contains only 30 questions. It is feasible and well worth reporting Pass@32 or Pass@64 on this dataset. Without large k results, the claim that CER preserves exploration remains insufficiently supported.
> >
> > ---
> >
> > **W3: Paper organization**
> >
> > Your rebuttal provides more convincing conceptual arguments for TECA, but these discussions should appear in the main text, not hidden in the rebuttal. Currently, Section 3 devotes an entire page to re-introducing GRPO and basic entropy definitions, which is already well-known.
> >
> > Moreover, the roles of Figure 1 and Figure 6 remain unclear. They appear highly redundant and do not introduce distinct insights. It's not a terrible mistake, however, given that the method’s novelty is moderate, it is worth questioning whether, after removing repeated formulas and nearly identical example figures, the actual contribution can substantively fill 10(or just 9) pages.
> >
> > ---
> >
> > Reference
> >
> > [1] Rewarding the Unlikely: Lifting GRPO Beyond Distribution Sharpening, He et al., 2025.

---

> > > ### Author Response · Authors · 2025-12-01
> > > **(1/2)**
> > >
> > > Thanks for your prompt responses and constructive comments.
> > >
> > > >  **W.1:** The authors provided no prior literature or reported experiment supporting this statement. If only the two models tested in the paper exhibit the “overthinking” phenomenon, then the motivation of studying this phenomenon becomes unclear. Overthinking is indeed an important behavioral pattern, but before proposing a mitigation method, one must rigorously establish which models exhibit it.
> > >
> > > **Re:** We thank the reviewer for raising this important point and appreciate the opportunity to provide a clearer and more rigorous justification.
> > >
> > > We acknowledge that our previous wording was unclear and may have sounded arbitrary. Below we provide the missing evidence and references supporting our claim.
> > >
> > > **1. Evidence of Overthinking in LLMs**
> > >
> > > Overthinking in LLMs is well-documented in prior literature, with two main perspectives used to measure it:
> > >
> > > - **Length-based**: Overthinking is identified when LLMs generate excessively long or detailed reasoning without improving accuracy for simple problems[1–3]. A formal definition provided by [6]:
> > >
> > >   > *Overthinking is the generation of additional thought tokens that do not contribute to a performance gain.*
> > >
> > >   Thus, length itself is an established quantitative indicator of overthinking.
> > >
> > > - **Efficiency-based**: Metrics like **Accuracy Efficiency Score (AES)** [4] and **Token Efficiency (TE)** [5] assess the trade-off between reasoning length and accuracy. *Lower values indicate overthinking*. The calculation of AES and TE are as follows:
> > >
> > >   **(1) Accuracy Efficiency Score (AES) [4]:**
> > >
> > >   ​    Proposed by O1-Pruner, AES evaluates the trade-off between improving accuracy and reducing computational cost. It is calculated by weighting and summing the model’s solution length and accuracy. Defining $\Delta \text{Length} = \frac{\text{Length}\_{baseline}-\text{Length}\_{model}}{\text{Length}\_{baseline}}$ and $\Delta \text{Acc} = \frac{\text{Acc}\_{model}-\text{Acc}\_{baseline}}{\text{Acc}\_{baseline}}$, the AES is computed as:
> > >   $$
> > >   \text{AES} = \begin{cases}
> > >       \alpha \cdot \Delta\text{Length} + \beta \cdot |\Delta\text{Acc}|, & \text{if } \Delta\text{Acc} \ge 0 \\\
> > >       \alpha \cdot \Delta\text{Length} - \gamma \cdot |\Delta\text{Acc}|, & \text{if } \Delta\text{Acc} < 0
> > >       \end{cases}
> > >   $$
> > >   Here, the baseline refers to the corresponding non-thinking model. We follow the O1-Pruner setting with $\alpha = 1, \beta = 3, \gamma = 5$. A higher AES indicates a better balance between precision and efficiency.
> > >
> > >   **(2) Token Efficiency (TE) [5]:**
> > >
> > >   To evaluate the reasoning accuracy relative to conciseness, [5] proposed the Token Efficiency metric, defined as:
> > >   $$
> > >   \text{TE} = \frac{A}{\sqrt{L}}
> > >   $$
> > >   Here, $A$ is the accuracy and $L$ is the response length. A higher TE indicates superior information density.
> > >
> > > **2. Overthinking in R1-distill-Qwen2.5-7B**
> > >
> > > To investigate overthinking issue in the R1-distill-Qwen2.5-7B model, we compare it with its base model (Qwen2.5-7B) and another variant, Qwen3-4B (with and without thinking mode).
> > >
> > > **(a) Length-based evidence**
> > >
> > > |                           | **gsm8k/acc** | **gsm8k/len** |
> > > | ------------------------- | ------------- | ------------- |
> > > | Qwen2.5-7B                | 86.27         | 315.42        |
> > > | **R1-distill-Qwen2.5-7B** | **56.48**     | **525.44**    |
> > > | Qwen3-4B w/o thinking     | 86.50         | 260.96        |
> > > | **Qwen3-4B w thinking**   | **92.80**     | **1348.59**   |
> > >
> > > **Observation**: R1-distill-Qwen2.5-7B generates significantly longer reasoning but without performance improvement, indicating overthinking.
> > >
> > > **(b) Efficiency-based evidence**
> > >
> > > |                           | **gsm8k/AES** | **gsm8k/TE** |
> > > | ------------------------- | ------------- | ------------ |
> > > | Qwen2.5-7B                | 0             | 4.86         |
> > > | **R1-distill-Qwen2.5-7B** | **-2.39**     | **2.46**     |
> > > | Qwen3-4B w/o thinking     | 0             | 5.35         |
> > > | **Qwen3-4B w thinking**   | **-3.95**     | **2.53**     |
> > >
> > > **Observation**: R1-distill-Qwen2.5-7B shows negative AES and lower TE, indicating less efficient reasoning, but it is less severe compared to Qwen3-4B with thinking mode.
> > >
> > > **Conclusion**
> > >
> > > Both length-based and efficiency-based analyses confirm that R1-distill-Qwen2.5-7B exhibits overthinking, but to a lesser extent than Qwen3-4B with thinking mode. This supports our original claim that **R1-distill-Qwen2.5-7B does experience overthinking, though not as severely**.

---

> > > > ### Author Response · Authors · 2025-12-01
> > > > **(2/2)**
> > > >
> > > > > **W.2:** You need to clarify how length is aggregated across k samples in the Pass@k evaluation in the table (e.g., mean or minimum among k, etc.).
> > > > > In addition, Pass@8 is far too small to draw meaningful conclusions. As shown in He et al. (2025) [1], Pass@k becomes informative when k ≥ 32, especially for math reasoning tasks. I fully understand that running large k sampling on models is expensive, but AIME24 contains only 30 questions. It is feasible and well worth reporting Pass@32 or Pass@64 on this dataset. Without large k results, the claim that CER preserves exploration remains insufficiently supported.
> > > >
> > > > **Re:** We thank the reviewer for pointing out these important issues and for the constructive suggestions.
> > > >
> > > > **1. Clarification on Length Aggregation**
> > > >
> > > > We confirm that the *length* reported in the Pass@k evaluation is computed as the **average number of generated tokens across the k samples** (i.e., *avg@k*).
> > > >
> > > > **2. Evaluation with Larger k**
> > > >
> > > > Following your recommendation, we conducted additional experiments on the GSM8K and AIME24 dataset with **Pass@32** and **Pass@64**. As shown in the table below, **CER continues to preserve exploration while significantly reducing overthinking**, even under larger sample sizes. These results further support the robustness of CER beyond the original Pass@8 setting.
> > > >
> > > > |                   | gsm8k/acc | gsm8k/len | aime24/acc | aime24/len |
> > > > | ----------------- | :-------: | :-------: | :--------: | :--------: |
> > > > | 4B-origin-pass@32 |   97.50   |  1357.55  |   83.33    |  11270.63  |
> > > > | 4B-CER-pass@32    |   97.19   |  567.26   |   86.67    |  9158.83   |
> > > > | 4B-origin-pass@64 |   97.80   |  1357.17  |   86.67    |  11294.85  |
> > > > | 4B-CER-pass@64    |   97.42   |  568.77   |   83.33    |  9039.03   |
> > > >
> > > > These new results further strengthen our claim that CER preserves necessary exploration while reducing excessive reasoning length, demonstrating robust effectiveness across different settings, even at higher k values as recommended.
> > > >
> > > > &ensp;
> > > >
> > > > > **W.3:** Your rebuttal provides more convincing conceptual arguments for TECA, but these discussions should appear in the main text, not hidden in the rebuttal. Currently, Section 3 devotes an entire page to re-introducing GRPO and basic entropy definitions, which is already well-known.
> > > > > Moreover, the roles of Figure 1 and Figure 6 remain unclear. They appear highly redundant and do not introduce distinct insights. It's not a terrible mistake, however, given that the method’s novelty is moderate, it is worth questioning whether, after removing repeated formulas and nearly identical example figures, the actual contribution can substantively fill 10(or just 9) pages.
> > > >
> > > > **Re:** Thank you for your thoughtful feedback.
> > > >
> > > > We would like to clarify that the conceptual arguments for TECA are not intended to be hidden in the rebuttal. We appreciate the reviewer’s suggestion to make these arguments more explicit, and we will incorporate a clearer and more solid discussion of the TECA mechanism in the main text of the final version, as recommended.
> > > >
> > > > Regarding Figure 1 and Figure 6, we included Figure 1 to provide an intuitive example of the overthinking issue, while Figure 6 demonstrates a case where our method effectively mitigates overthinking in LRMs using the CER mechanism. To streamline the presentation, we will move Figure 6 to the appendix in final version.
> > > >
> > > > Regarding the contributions of our work, we firmly believe they are substantial. The proposed **TECA metric** offers a novel mean of quantifying exploratory behavior throughout the reasoning process, and the **CER mechanism** leverages TECA to guide models toward the correct solution while avoiding overthinking in LRMs. Together, these contributions enable our method to effectively mitigate the overthinking issue in LLMs **without** compromising problem-solving ability. In the final version, we will incorporate the additional details discussed in the rebuttal to provide a more comprehensive and polished presentation of these contributions, which will appropriately and substantively fill  9–10 pages.
> > > >
> > > > We also believe that the value of a research contribution is not solely determined by the page count, but rather by the novelty and soundness of the ideas presented.
> > > >
> > > > ------
> > > >
> > > > **References:**
> > > >
> > > > [1] When More is Less: Understanding Chain-of-Thought Length in LLMs
> > > >
> > > > [2] Stop Overthinking: A Survey on Efficient Reasoning for Large Language Models
> > > >
> > > > [3] Do NOT Think That Much for 2+3=? On the Overthinking of o1-Like LLMs
> > > >
> > > > [4] O1-Pruner: Length-Harmonizing Fine-Tuning for O1-Like Reasoning Pruning
> > > >
> > > > [5] Reconsidering Overthinking: Penalizing Internal and External Redundancy in CoT Reasoning
> > > >
> > > > [6] Do LLMs Really Need 10+ Thoughts for “Find the Time 1000 Days Later”? Towards Structural Understanding of LLM Overthinking

---

### Official Review · Reviewer_bUgf · 2025-11-05

**Soundness:** 3
**Presentation:** 3
**Contribution:** 3
**Rating:** 6
**Confidence:** 4

**Summary:**

The paper looks at the problem of overthinking in chain-of-thought reasoning efficiency by regulating token-level entropy. They propose a framework with two key components: TECA, which measures cumulative uncertainty, and CER, which penalizes excessive entropy to encourage concise reasoning. They explore (1) reasoning dynamics, where they analyze entropy changes during exploration and determination, and (2) RL integration, where they embed CER into GRPO to control reasoning depth. They also extend their work to mathematical reasoning benchmarks and show consistent reductions in reasoning length with minimal accuracy loss.

**Strengths:**

The idea of regulating token-level entropy to mitigate overthinking in LLM reasoning is both novel and well-motivated. Most existing work focuses on increasing reasoning accuracy through longer CoT generation or reinforcement learning rewards, but these approaches cannot adaptively control reasoning depth or prevent redundant exploration. This would limit their efficiency and interpretability. This paper introduces a method that improves reasoning efficiency without sacrificing accuracy. This makes the contribution practically impactful and valuable to the community.


The workflow is well-structured, as it integrates entropy-based reasoning control with GRPO to make the reasoning process adaptive and self-regulated. This method allows LLMs to decide when to stop exploring.

The experiments are extensive, with detailed analysis of the results. These experiments validate the effectiveness across different model sizes and reasoning complexities.

**Weaknesses:**

i. The experiments are now all on math tasks. In math tasks the reasoning process usually follows a clear logical progression: i. the model explores multiple computational paths ii. converges to a definitive conclusion. In this setting, token-level entropy does effectively reflect the model’s degree of cognitive divergence. The cumulative entropy can capture the transition from exploration to determination relatively easily. The assumption may not work well on open-ended and/or textual tasks. E.g., QA tasks lack a clear boundary between exploration and determination. The model simultaneously retrieves facts, interprets semantics, and formulates answers. This could cause the TECA to mix different cognitive phases and produce noisy signals. For open-ended or multi-answer tasks, the evaluation signals are ambiguous and entropy penalties may prematurely constrain generation. I would suggest that the entropy-based works dig deeper into diverse tasks.

ii. The contribution of each component is not fully isolated. Need an ablation study.

iii. A stability and sensitivity study is needed, as the entropy-based rewards may be sensitive to temperature, model calibration, and/or decoding policy.

iv. I would suggest adding a clock time report or computational cost, although the reasoning length is reduced.

**Questions:**

See above.

---

> ### Author Response · Authors · 2025-11-20
> **(1/2)**
>
> Thank you for your valuable comments. After carefully considering your feedback, we have decided to provide the following corresponding explanations.
>
> > **Weakness 1:** The experiments are now all on math tasks. In math tasks the reasoning process usually follows a clear logical progression: i. the model explores multiple computational paths ii. converges to a definitive conclusion. In this setting, token-level entropy does effectively reflect the model’s degree of cognitive divergence. The cumulative entropy can capture the transition from exploration to determination relatively easily. The assumption may not work well on open-ended and/or textual tasks. E.g., QA tasks lack a clear boundary between exploration and determination. The model simultaneously retrieves facts, interprets semantics, and formulates answers. This could cause the TECA to mix different cognitive phases and produce noisy signals. For open-ended or multi-answer tasks, the evaluation signals are ambiguous and entropy penalties may prematurely constrain generation. I would suggest that the entropy-based works dig deeper into diverse tasks.
>
> **Re:** Thank you for pointing out this limitation. It is correct that our current experiments focus mainly on mathematical reasoning benchmarks. This choice is motivated by the following considerations:
>
> First, the overthinking phenomenon is most prominent in LRMs—particularly Long CoT models [1, 2]. Accordingly, our base models are all open-source Long CoT models. Overthinking issues also tend to arise in tasks requiring strong multi-step reasoning, which are predominantly mathematical benchmarks.
>
> Second, our evaluation follows the setups used in prior work. Existing studies addressing overthinking or aiming to improve LRMs’ reasoning capability/efficiency also primarily evaluate on mathematical datasets [3–9]. To ensure comparability and consistency with these methods, we adopted the same evaluation protocol.
>
> That said, we appreciate your suggestion regarding broader task types. To examine this, we additionally conducted experiments on a non-mathematical dataset, BoolQ [10], which differs substantially from mathematical benchmarks. The model trained with CER achieved a **45.20% reduction** in output length while maintaining nearly the same accuracy as the base model. This result suggests that CER can extend beyond mathematical reasoning settings and remains effective on textual comprehension tasks.
>
> | model        | accuracy | length |
> | ------------ | :------: | :----: |
> | qwen3-4B     |  87.40   | 403.43 |
> | qwen3-4B-CER |  87.19   | 221.08 |
>
> > **Weakness 2:** The contribution of each component is not fully isolated. Need an ablation study.
>
> **Re:** Thank you for this helpful suggestion. We have added an ablation study to more clearly isolate the contributions of each component.
>
> The CER framework consists of two key modules: (1) the TECA reward and (2) the SRM. Because the SRM is defined based on the TECA reward, the ablation mainly examines the effect of removing SRM and the effect of removing both TECA and SRM.
>
> The results show that removing both TECA and SRM yields higher accuracy than full CER, but also leads to substantially longer responses. When only the SRM is removed, we observe a reduction in output length compared with the original model; however, accuracy on easy problems drops noticeably, while accuracy on hard problems shows a slight improvement. The corresponding change in length is also smaller than that achieved by full CER.
>
> We consider the minor fluctuations on hard problems to be expected. This is related to the training strategy of CER: the model is encouraged to determine an answer and maintain confidence once it reaches a stable conclusion. This mechanism can occasionally lead the model to converge to an incorrect answer, which in turn introduces small variations in accuracy, particularly on more challenging tasks. Nonetheless, this effect is limited and does not alter the overall trends. The detailed results are presented in the table below:
>
> |                     | GSM8K/acc | GSM8K/len | AIME24/acc | AIME24/len |
> | :-----------------: | :-------: | :-------: | :--------: | :--------: |
> |    qwen3-4B-CER     |   94.09   |  391.08   |   61.11    |  9215.77   |
> | qwen3-4B-remove-SRM |   91.89   |  542.61   |   62.22    |  8750.14   |
> |  qwen3-4B-acc-only  |   95.22   |  887.07   |   64.44    |  10953.92  |

---

> ### Author Response · Authors · 2025-11-20
> **(2/2)**
>
> > **Weakness 3:** A stability and sensitivity study is needed, as the entropy-based rewards may be sensitive to temperature, model calibration, and/or decoding policy.
>
> **Re:** Thank you for highlighting this issue. We agree that examining stability and sensitivity is important, especially since entropy-based rewards may be influenced by temperature settings, calibration, and decoding policies. We therefore conducted a set of experiments to assess the sensitivity of CER with respect to the rollout temperature.
>
> The results are summarized below. Adjusting the rollout temperature directly affects the degree of exploration during training: higher temperatures yield more stochastic outputs, which expand the exploration space and lead to correspondingly larger parameter update steps. As shown in our experiments, for **easy problems**, increasing the temperature generally enhances the length reduction effect, while also yielding modest improvements in accuracy.
>
> For **hard problems**, the length reduction is less sensitive to temperature. This is expected, as solving these tasks inherently requires substantial reflection and exploration regardless of the sampling stochasticity. Interestingly, the accuracy at **temperature = 0.8** is noticeably higher than at other temperatures. We attribute this to the fact that lower output randomness slows the model’s acquisition of the targeted reasoning pattern. At this stage, the model has not fully learned the “explore briefly, then decide” behavior, which results in higher accuracy but also significantly longer responses.
>
> The detailed quantitative results are provided in the table below.
>
> | temperature | GSM8K/acc | GSM8K/len | AIME24/acc | AIME24/len |
> | ----------- | --------- | --------- | ---------- | ---------- |
> | 1.5         | 94.09     | 391.08    | 61.11      | 9215.77    |
> | 1.2         | 93.71     | 456.44    | 61.11      | 8703.79    |
> | 0.8         | 93.71     | 606.27    | 66.67      | 9646.43    |
>
> > **Weakness 4:** I would suggest adding a clock time report or computational cost, although the reasoning length is reduced.
>
> **Re:** We appreciate the reviewer’s suggestion. Below we provide a detailed report on the computational cost of CER from both the training and inference perspectives.
>
> **Inference cost.**
> Under identical conditions, the model trained with CER achieves substantially faster inference than the original baseline. On the GSM8K test set, the original model requires **8.242 s/problem**, whereas the CER model requires **3.208 s/problem**, achieving nearly a **3× speedup**. This improvement aligns with the reduced reasoning length induced by CER.
>
> **Training cost.**
> CER does introduce additional training overhead, primarily in GPU memory consumption. Our implementation uses *verl* as the training framework. To support CER, we modified its data flow so that entropy values computed during training can be retained for reward computation. In the standard pipeline, entropy is freed immediately after log-probability computation; however, CER requires preserving this information until the reward is computed. As a result, memory usage is higher during the segment between log-prob computation and reward calculation.
>
> &nbsp;
>
> **Reference:**
>
> [1] Do NOT Think That Much for 2+3=? On the Overthinking of o1-Like LLMs
>
> [2] Stop Overthinking: A Survey on Efficient Reasoning for Large Language Models
>
> [3] Sample More to Think Less:  Group Filtered Policy Optimization for Concise Reasoning
>
> [4] Wait, We Don’t Need to “Wait”!  Removing Thinking Tokens Improves Reasoning Efficiency
>
> [5] When More is Less: Understanding Chain-of-Thought Length  in LLMs
>
> [6] Efficient Reasoning for Large Reasoning Language Models via Certainty-Guided Reflection Suppression
>
> [7] Pruning Long Chain-of-Thought of Large Reasoning Models via Small-Scale Preference Optimization
>
> [8] Beyond the 80/20 Rule: High-Entropy Minority Tokens Drive Effective Reinforcement Learning for LLM Reasoning
>
> [9] The Entropy Mechanism of Reinforcement Learning for Reasoning Language Models
>
> [10] BoolQ: Exploring the Surprising Difficulty of Natural Yes/No Questions

---

### Author Response · Authors · 2025-11-24

Dear reviewers:

We thank the reviewers for the constructive feedback. We have carefully addressed all reviewer comments in our rebuttal, and we have also updated the corresponding parts of the manuscript in the latest version to reflect these clarifications and improvements. If there are any additional questions or concerns, we would be happy to further elaborate and provide any supplementary explanations needed.

Best regards

Authors

---

### Author Response · Authors · 2025-12-01
**Global Response(1/2)**

Dear PCs, SACs, ACs, and all of our reviewers:

We are grateful for the constructive comments and detailed reviews provided by the reviewers. These insights have been essential in refining our manuscript, and we have carefully revised the text to incorporate your suggestions and address the questions raised.

We are encouraged by your recognition of our paper's strengths:

- Reviewers bUgf, 6psw and oyuY highlighted that "**TECA provides a novel and interpretable way to monitor when a model is overthinking by measuring uncertainty accumulation during reasoning**" and "**shows reasonable empirical correlation with response length**".
- Reviewers oyuY and nNHM recognized the practical value of CER, due to "**on arithmetic and mathematics benchmarks the method reducing average reasoning length by up to 71 % while maintaining solving performance**".
- Reviewer bUgf praised that "**the workflow is well-structured, as it integrates entropy-based reasoning control with GRPO to make the reasoning process adaptive and self-regulated. This method allows LLMs to decide when to stop exploring.**"

Regarding the **conceptual analysis** raised by Reviewer **6psw**, we clarified the **causal diagnostic of TECA** and conducted **pro and con intervention study** as evidence. Furthermore, we provided detailed **theoretical** analysis and mathematical proofs to demonstrate the **limitations** of simpler uncertainty surrogates (such as logit margin and variance). This establishes TECA as a more **rigorous** metric for evaluating overthinking. We also **clarified the theoretical motivation of CER reward construction** and gave a sample proving that rewarding TECA only on correct samples will not encourage lucky guesses.

We have also addressed other primary concerns by reviewers during discussion period:

- To address the **concern of generalization ability** on non-mathematics benchmarks(Reviewers **bUgf**, **6psw** and **oyuY**), we clarified that our evaluation followed the setups used in prior work addressing overthinking or aiming to improve LRMs’ reasoning capability/efficiency[1-6], so we mainly choose mathematics benchmarks. Furthermore, as suggested, we conducted experiments on the QA task mentioned by Reviewer **bUgf** and results validated the generalization ability of CER(**45.20% length reduction** on BoolQ).

- In response to the **suggestions of ablation study and sensitive study**(Reviewers **bUgf** and **oyuY**), we supplemented the module ablation study with scenes of TECA + SRM, TECA and no TECA, and temperature sensitivity during rollout as sensitive study with temperature 1.5, 1.2, 0.8. The observed performance variation validated that TECA + SRM with temperature 1.5 **achieved the best performance and still remained robust**.

|                     | GSM8K/acc | GSM8K/len  | AIME24/acc | AIME24/len  |
| :-----------------: | :-------: | :--------: | :--------: | :---------: |
|  **qwen3-4B-CER**   | **94.09** | **391.08** | **61.11**  | **9215.77** |
| qwen3-4B-remove-SRM |   91.89   |   542.61   |   62.22    |   8750.14   |
|  qwen3-4B-no-TECA   |   95.22   |   887.07   |   64.44    |  10953.92   |
| **temperature 1.5** | **94.09** | **391.08** | **61.11**  | **9215.77** |
|   temperature 1.2   |   93.71   |   456.44   |   61.11    |   8703.79   |
|   temperature 0.8   |   93.71   |   606.27   |   66.67    |   9646.43   |


- To address the questions **whether CER will negatively impact the model exploration behavior** such as truncation of certain important reasoning exploration steps or under-explore on complex problems(Reviewers **nNHM**, **oyuY** and **6psw**), we supported the pass@k results and TECA curve analysis on complex problems.  Pass@k results still remaining robust demonstrated that **CER suppresses useless branches instead of simply cutting off exploration globally**. The TECA curve provided the evidence that the **model’s exploration capability is not weakened** after training.

|                   | gsm8k/acc | gsm8k/len | aime24/acc | aime24/len |
| :---------------: | :-------: | :-------: | :--------: | :--------: |
| 4B-origin-pass@4  |   96.29   |  1373.61  |   73.33    |  11199.52  |
|   4B-CER-pass@4   |   95.98   |  561.93   |   76.67    |  9325.61   |
| 4B-origin-pass@8  |   96.89   |  1359.78  |   83.33    |  11328.77  |
|   4B-CER-pass@8   |   96.36   |  571.70   |   83.33    |  9272.23   |
| 4B-origin-pass@32 |   97.50   |  1357.55  |   83.33    |  11270.63  |
|  4B-CER-pass@32   |   97.19   |  567.26   |   86.67    |  9158.83   |
| 4B-origin-pass@64 |   97.80   |  1357.17  |   86.67    |  11294.85  |
|  4B-CER-pass@64   |   97.42   |  568.77   |   83.33    |  9039.03   |

Finally, we sincerely thank the reviewers for their valuable suggestions. We believe that these additional results and clarifications make CER a clearer, more solid, and more comprehensive solution for mitigating overthinking.

Thanks again for your time and efforts.

Best regards,

The Authors

---

> ### Author Response · Authors · 2025-12-01
> **Global Response(2/2)**
>
> References:
>
> [1] Sample More to Think Less:  Group Filtered Policy Optimization for Concise Reasoning
>
> [2] Wait, We Don’t Need to “Wait”!  Removing Thinking Tokens Improves Reasoning Efficiency
>
> [3] When More is Less: Understanding Chain-of-Thought Length  in LLMs
>
> [4] Efficient Reasoning for Large Reasoning Language Models via Certainty-Guided Reflection Suppression
>
> [5] Pruning Long Chain-of-Thought of Large Reasoning Models via Small-Scale Preference Optimization
>
> [6] Beyond the 80/20 Rule: High-Entropy Minority Tokens Drive Effective Reinforcement Learning for LLM Reasoning

---

### Meta-Review · Area_Chair_2esi · 2026-01-05

**Summary:**

This paper investigates “overthinking” in long chain-of-thought LLMs and proposes Token Entropy Cumulative Average (TECA) as a diagnostic signal, based on the observation that cumulative token entropy tends to keep rising when the model continues to explore/rethink. Building on this, the authors propose CER, which uses TECA inside an RL reward to encourage the model to “explore briefly, then decide,” aiming to reduce unnecessary reasoning length while preserving accuracy. Reviews and discussion were mixed: some reviewers found TECA an interesting and interpretable observation and agreed the method can reduce average length, while others were not convinced the proposed RL-based solution is sufficiently justified or clearly stronger than existing concise-reasoning / pruning approaches, and raised concerns on generality and conceptual depth.

A separate and important issue is writing quality and formating. The paper reads rushed and insufficiently polished in organization (e.g., Figures 3 & 4 being placed far from where they are referenced, and parts of the paper feeling redundant rather than focused on the core idea). More seriously, the authors **clearly altered the paper layout to be significantly wider than the official template**, which violates code of conduct.

After weighing the factors above, I recommend rejection.

**Reviewer Concerns:**

The rebuttal was extensive and addressed a number of factual concerns, including adding pass@k results (including larger k), additional ablations, and more discussion on computational cost and sensitivity to temperature. These updates partially addressed concerns raised by reviewers bUgf and oyuY regarding ablations, stability, and exploration suppression, and reviewer nNHM was largely satisfied after discussion and maintained a positive score.

However, several core concerns remain unresolved. Reviewer 6psw consistently argued that while TECA is an interesting correlation signal, the paper does not provide a fully convincing causal or theoretical justification that TECA is the right quantity to regulate, rather than a proxy that correlates with length. This concern persists even after the rebuttal, which relies heavily on indirect arguments and post-hoc reasoning rather than clean interventions or principled derivations. More importantly, multiple reviewers noted that the proposed solution—using TECA as an RL reward—shows limited or marginal gains compared to prior concise-reasoning or pruning-based methods, especially given the additional RL complexity. As pointed out by reviewer 6psw, the improvement over existing approaches is not clearly compelling enough to justify the added machinery.

In addition, concerns about generality remain. While the authors added BoolQ results and additional discussion, the evidence is still largely concentrated on math benchmarks and specific long-CoT models, and the broader claims about overthinking and adaptive reasoning feel overstated relative to the demonstrated scope. Finally, there are serious presentation and professionalism issues. Reviewer 6psw noted that the paper appears rushed and insufficiently polished.

**Reviewer Scores:**

Reviewer bUgf initially rated the paper slightly above the acceptance threshold and would likely remain cautiously positive after the rebuttal, though still with reservations about generality. Reviewer nNHM gave a strong accept score and explicitly maintained their rating after discussion. Reviewer oyuY was marginally negative and might move slightly upward given the additional experiments, but likely still around the borderline. In contrast, reviewer 6psw was consistently negative throughout the discussion and explicitly stated that the rebuttal did not resolve the core concerns; their score would not increase and remains a strong reject. Overall, the scores remain mixed, with one strong accept, one mild accept, one borderline, and one strong reject, and the unresolved conceptual and presentation issues dominate the final recommendation.

---

### Decision · Program_Chairs · 2026-01-26

Reject